# INCENTIVIZING AGENTIC REASONING IN LLM JUDGES VIA TOOL-INTEGRATED REINFORCEMENT LEARNING

**Ran Xu**[1,2]*, **Jingjing Chen**[2], **Jiayu Ye**[2], **Yu Wu**[2], **Jun Yan**[3], **Carl Yang**[1], **Hongkun Yu**[2]
[1]Emory University, [2]Google, [3]Google Cloud AI Research

## ABSTRACT

Large Language Models (LLMs) are widely used as judges to evaluate response quality, providing a scalable alternative to human evaluation. However, most LLM judges operate solely on intrinsic text-based reasoning, limiting their ability to verify complex constraints or perform accurate computation. Motivated by the success of tool-integrated reasoning (TIR) in numerous tasks, we propose `TIR-Judge`, an end-to-end RL framework for training LLM judges that integrates a Python executor for precise evaluation. `TIR-Judge` is built on three principles: (i) diverse training across verifiable and non-verifiable domains, (ii) flexible judgment formats (pointwise, pairwise, listwise), and (iii) iterative RL that enables bootstrapping directly from a base model without distillation. On seven public benchmarks, `TIR-Judge` surpasses strong reasoning-based judges by up to 6.4% (pointwise) and 7.7% (pairwise), and achieves listwise performance comparable to Claude-Opus-4 despite having only 8B parameters. Remarkably, `TIR-Judge`-Zero—trained entirely without distillation—matches the performance of the distilled variants, showing that tool-augmented judges can self-improve through reinforcement learning alone.

## 1 INTRODUCTION

Large Language Model (LLM)-based judges are emerging as a critical component in the LLM ecosystem, typically used with scoring and ranking model outputs. This evaluation capability is essential at multiple stages of LLM development: during post-training, judges provide preference signals for alignment (Chen et al., 2025a; Whitehouse et al., 2025); at inference time, judges verify and select responses through best-of-N decoding (Huang et al., 2025a); and during evaluation, judges deliver reliable assessments without manual human assessment (Li et al., 2024a). Thus, training accurate LLM-based judges is of great importance for building powerful language models.

Classical evaluation with reward models often outputs scores directly, which cannot fully harvest the inherent reasoning capability of LLM. Recent progress in generative reward modeling (Zhang et al., 2025; Zhao et al., 2025) and reinforcement learning equips judges with thinking before producing final predictions (Chen et al., 2025b; Whitehouse et al., 2025; Guo et al., 2025b; Hong et al., 2025). While these approaches enhance judge quality by equipping LLMs with long chains of textual reasoning traces, they remain inherently limited in scenarios that require precise computation or symbolic reasoning – capabilities that are much more challenging for text-only models (Mirzadeh et al., 2025).

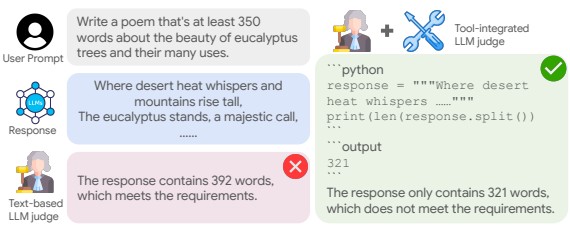

Figure 1: An example of LLM judge augmented with code execution, enabling precise judgments.

Recent advances in LLM tool-use provide a promising avenue to overcome the limitations of text-only judges (Chen et al., 2023; Gao et al., 2023). By granting access to executable interfaces for enumeration, verification, and computation, tools enable exact validation of reasoning steps rather than relying on potentially error-prone text-based inference. For example, code execution

---

*Work done during an internship at Google. Corresponding Email: ran.xu@emory.edu.

can automatically *verify outputs on certain instructions* (Zhou et al., 2023) (as shown in Figure 1) or *check intermediate calculations* in math reasoning (Lu et al., 2025). Early attempts have also explored equipping LLM judges with tool-use abilities (Peng et al., 2025; Findeis et al., 2025; Li et al., 2024b; Agarwal et al., 2025), but these approaches reveal two major limitations. (i) *Inference-time restriction*: most methods integrate tool-use only at the inference stage, preventing deeper integration between reasoning processes and tool execution. (ii) *Narrow task coverage*: many are tailored to specific domains or specialized task types, which limits their applicability in general-purpose judging scenarios. These gaps highlight the need for robust judges that tightly couple reasoning with tool execution and be optimized end-to-end.

Motivated by these challenges, our goal is to develop an LLM judge that can reliably integrate reasoning with code interpreter execution. Incorporating tool-integrated reasoning (TIR) (Feng et al., 2025; Li et al., 2025a; Lin & Xu, 2025), we propose `TIR-Judge`, a framework that leverages reinforcement learning (RL) to teach models to generate code, execute it with interpreters, and iteratively refine their reasoning based on the resulting outputs. By reinforcing this cycle of reasoning and tool-use, `TIR-Judge` equips LLM judges with the ability at the training time to deliver more accurate and verifiable evaluations across diverse tasks.

Then, to fully unleash the potential of RL for `TIR-Judge`, we introduce several key design choices. (i) *Task diversity*: To balance between different tasks, we construct training prompts spanning both verifiable domains (e.g., competitive programming, mathematical reasoning) and non-verifiable domains (e.g., dialogue, safety, general coding), allowing the model to learn when tool invocation is beneficial and when pure reasoning suffices. (ii) *Judgment flexibility*: To accommodate to different input/output formats, we diversify the evaluation tasks to cover pointwise, pairwise, and listwise ranking, ensuring broad applicability across practical use cases. (iii) *Data efficiency*: unlike prior methods that rely on distillation as cold-start for RL (Chen et al., 2025b; Hong et al., 2025), we demonstrate that `TIR-Judge` can bootstrap from the initial checkpoint. Specifically, `TIR-Judge`-Zero trains purely with iterative reinforcement learning for achieving self-improvement, while `TIR-Judge`-Distill provides an optional variant using a small amount of distillation data.

Our contribution can be summarized as follows:

- We introduce `TIR-Judge`, a tool-integrated framework for training LLM-based judges with end-to-end multi-turn reinforcement learning. To the best of our knowledge, this is the first approach that jointly optimizes reasoning and tool-use for training LLM-based judges via RL.

- We design several key strategies to fully exploit the power of reinforcement learning, including *task diversification* across verifiable and non-verifiable domains, *flexible judgment formats* (pointwise, pairwise, listwise), as well as an *iterative RL scheme* that enables self-improvement in tool use even without distillation.

- We evaluate `TIR-Judge` on seven public benchmarks covering diverse tasks and input formats. `TIR-Judge` consistently outperforms strong reasoning-based judges, achieving gains of up to 6.4% (pointwise) and 7.7% (pairwise). Moreover, `TIR-Judge` shows strong parameter efficiency: With only 8B parameters, it surpasses the 32B reasoning reward models on the PPE dataset, and reaches 96% of the performance of Claude-Opus-4 in the listwise setting in RewardBench 2. Interestingly, `TIR-Judge`-Zero, the judge trained without any distillation, achieves a 1.2% gain over its distilled counterpart at 4B scale, highlighting the power of RL to bootstrap reasoning and tool-use capabilities.

## 2 RELATED WORKS

**Reasoning-Enhanced Reward and Judge Models.** A growing line of work strengthens reward models (RMs) and judges by explicitly training them to *reason* before scoring. Generative Verifiers (Zhang et al., 2025) treat verification as next-token prediction with chain-of-thought, improving math and algorithmic tasks. Other methods enhance judgment by generating critiques (Ankner et al., 2024; Yu et al., 2025c;b; Wang et al., 2025a), using multi-round preference optimization (Wang et al., 2024b), or planning evaluations before issuing a decision (Saha et al., 2025). Liu et al. (2025d) study how to allocate compute and structure critiques for reliability. More recently, RL-based judges (Chen et al., 2025a;b; Whitehouse et al., 2025; Guo et al., 2025b; Hong et al., 2025) incentivize longer, higher-quality reasoning and reduce bias across pointwise and pairwise settings, Khalifa et al. (2025);

Zhao et al. (2025) leverage thinking traces to improve process reward models. However, these remain confined to text-only reasoning and often emphasize pairwise judgment.

**Tool-Assisted Reward and Judge Models.** Another line of work augments judges with external tools. Li et al. (2024b) incorporate verifiable signals alongside preference data for judge training, though primarily within tool-use scenarios. Zhuge et al. (2025) evaluate agentic judge capabilities in agent settings, and Peng et al. (2025) integrate human preferences with correctness checks to construct more reliable rewards. Findeis et al. (2025) show that external tools (e.g., code execution, search) can improve annotations, observing gains are task-dependent. Yet, most works rely on *prompted* tool use rather than training judges to *learn when and how* to call tools and incorporate their outputs.

**Reinforcement Learning for Tool-integrated Reasoning.** RL has recently been applied to tool-integrated reasoning (TIR). Feng et al. (2025); Bai et al. (2025); Li et al. (2025a) train LLMs to interleave reasoning with code execution, improving math and programming. Jin et al. (2025); Song et al. (2025) extend this to web search, while others study reward design (Dong et al., 2025; Wang et al., 2025b) or provide theoretical analyses (Lin & Xu, 2025).

## 3 PRELIMINARIES

**Problem Setup.** We consider the task of *LLM-based judgment*: given a user prompt $x \in \mathcal{X}$ and $n$ model-generated responses $\mathcal{Y} = \{y_1, y_2, \ldots, y_n\}$, the goal is to evaluate the quality of responses for the prompt. The judge model $J_\theta$ produces an evaluation output conditioned on $(x, \mathcal{Y})$. In this work, we consider three evaluation settings: (i) **Pointwise evaluation**: given $(x, y)$, the judge assigns a scalar score, $J_\theta(x, y) = s_\theta(x, y) \in \mathbb{R}$; (ii) **Pairwise evaluation**: given $(x, y_a, y_b)$, the judge selects the preferred response, $J_\theta(x, y_a, y_b) = \arg\max_{i \in \{a,b\}} s_\theta(x, y_i)$, where $s_\theta$ denotes a learned scoring function. This is also the most common evaluation setting; (iii) **Listwise evaluation**: given $(x, \mathcal{Y})$ with $n > 2$, the judge returns the index of the best response, $J_\theta(x, \mathcal{Y}) = \arg\max_{i \in \{1,\ldots,n\}} s_\theta(x, y_i)$. These settings unify a broad range of evaluations under a common framework[1].

**Tool-Augmented Judge.** We extend the judge with the ability to call an external *Python execution environment* $\mathcal{I}$. For the prompt $x \in \mathcal{X}$, At step $k$, the judgment trajectory $s_k$ is represented as $s_k = \{r_1, c_1, o_1, \ldots, r_k, c_k, o_k\}$, where $r_i$ is a natural language reasoning step, $c_i$ is a generated code, and $o_i = I(c_i)$ is the execution result of $c_i$ (Li et al., 2025a). The iterative process is defined as:

$$(r_k, c_k) \sim J(x \oplus s_{k-1}), \quad o_k = \mathcal{I}(c_k), \quad s_k = s_{k-1} \oplus r_k \oplus c_k \oplus o_k. \quad (1)$$

This cycle continues until the judge produces a final prediction $a_i \sim J(x \oplus s_T)$ with $T$ being the final step. Unlike traditional text-only reasoning, the trajectory now interleaves reasoning, code execution, and tool feedback, enabling the judge to ground its decision in verifiable evidence.

## 4 TRAINING TIR-JUDGE

We now describe the training procedure for TIR-Judge, which consists of four components: (1) data collection and filtering for RL, (2) the RL framework for training judges with integrated code execution tools, (3) reward design for RL, and (4) cold-start and iterative training strategies in RL. The overall framework of TIR-Judge is exhibited in Figure 2.

### 4.1 DATA COLLECTION AND FILTERING

High-quality training data is crucial for RL with tool-augmented judges. Since judgment requires both prompts and candidate responses, we curate a collection of (prompt, responses) tuples spanning multiple tasks. Our corpus integrates both human-annotated preference data and automatically generated synthetic pairs to ensure diversity and scalability.

**Real Preference Pairs.** We sample human-labeled preference pairs from a variety of domains: **general helpfulness** — *HelpSteer 3* (Wang et al., 2025c); **reasoning** — *UltraInteract* (Yuan et al., 2025), *S1* (Muennighoff et al., 2025); **coding** — *CodeRM* (Ma et al., 2025b); **instruction following**

---

[1]Note that in our work, the reference answer is *unseen* during evaluation, different from the *verification setting* (Liu et al., 2025a; Li et al., 2025b; Yan et al., 2025) where the reference answer is also a part of the input.

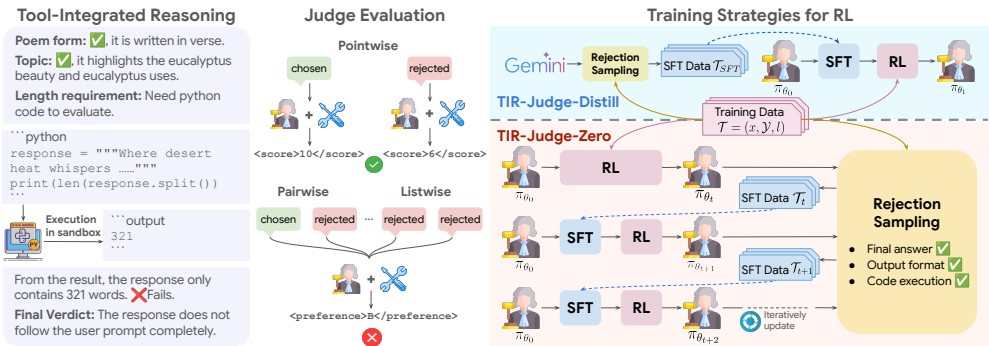

Figure 2: Overall framework of `TIR-Judge` variants. `TIR-Judge` natively supports tool use during judgment and is designed to handle diverse input formats.

**(IF)** — preference pairs from *Tulu 3* (Lambert et al., 2024); **safety** — *Safe-RLHF* (Dai et al., 2024). Each prompt is paired with one preferred (*chosen*) response and one or more *rejected* responses.

**Synthetic Preference Pairs.** Because reasoning preference data is often limited in scale, we augment the corpus with *synthetic* preference pairs generated from verifiable prompts. For each prompt, we sample responses from multiple open-source models, including *Qwen3-8B/14B* (Team, 2025), *Gemma-2-9B* (Team et al., 2024), and *Gemma-3-12B* (Team et al., 2025). The responses are automatically evaluated against verifiable functions (for IF tasks) or ground-truth solutions (for reasoning tasks) to form preference pairs. For **IF**, we use verifiable prompts from *Tulu-3* (Lambert et al., 2024), where correctness can be programmatically verified using lexical or structural constraints. For **reasoning**, we employ *MATH* (Hendrycks et al., 2021) and *DAPO-Math* (Yu et al., 2025a) for math domain and *WebInstruct* (Ma et al., 2025a), and *Loong* (Huang et al., 2025b) for general domain, both of which provide ground-truth solutions for exact verification.

In total, our dataset comprises approximately 26k preference pairs, including pointwise, pairwise, and listwise annotations, covering diverse domains such as helpfulness, reasoning, coding, safety, and verifiable instruction following. We apply strict 8-gram decontamination to eliminate any overlap between training prompts and evaluation benchmarks (Oren et al., 2024). This diverse mixture of data provides a strong foundation for training robust tool-augmented judges.

## 4.2 TOOL-INTEGRATED RL WITH VERIFIABLE REWARDS

**Overall Framework.** We adopt DAPO (Yu et al., 2025a), an improved variant of GRPO (Shao et al., 2024), for training the LLM judge $J$ parameterized by $\pi_\theta$. Given a prompt–answer pair $(q, a)$, we first sample a group of $G$ rollouts $\{s_i\}_{i=1}^G$ from the current policy $\pi_{\theta_{\text{old}}}$. Each rollout $s_i$ is assigned a scalar reward $R_i = R(s_i, a)$ with access to the oracle answer $a$. The policy $\pi_\theta$ is then updated with the following clipped policy gradient objective:

$$\mathcal{J}(\theta) = \mathbb{E}_{(q,a)\sim\mathcal{D}, \{s_i\}_{i=1}^G \sim \pi_{\theta_{\text{old}}}(\cdot|q)} \left[ \frac{1}{\sum_{i=1}^G |s_i|} \sum_{i=1}^G \sum_{t=1}^{|s_i|} \left( \min(r_{i,t}(\theta)\widehat{A}_{i,t}, \right. \right.$$

$$\left. \left. \text{clip}(r_{i,t}(\theta), 1-\varepsilon_{\text{low}}, 1+\varepsilon_{\text{high}})\widehat{A}_{i,t}) - \beta\mathbb{D}_{\text{KL}}(\pi_\theta\|\pi_{\text{ref}}) \right) \right] \quad \text{s.t. } 0 < |\{s_i : \texttt{is\_equivalent}(a, s_i)\}| < G$$

where $r_{i,t}(\theta) = \frac{\pi_\theta(s_{i,t}|q,s_{i,<t})}{\pi_{\theta_{\text{old}}}(s_{i,t}|q,s_{i,<t})}$ is the token-level weight, $\widehat{A}_{i,t} = \frac{R_i - \text{mean}(\{R_i\}_{i=1}^G)}{\text{std}(\{R_i\}_{i=1}^G)}$ is the advantage at the token level, and `is_equivalent` step filters out the prompts with accuracy equal to 1 and 0. The hyperparameters $\varepsilon_{\text{low}}$ and $\varepsilon_{\text{high}}$ control the clipping range for importance weights, while $\beta$ regulates the KL divergence penalty to stabilize training.

**Additional Designs.** Beyond standard RL training, we implement two enhancements to stabilize tool-augmented judgment: *(i) Error Message Processing.* We truncate the outputs from Interpreter $\mathcal{I}$ to only the final error line to avoid excessive context length while preserving useful feedback in $s_k$; *(ii) Sandbox Output Masking.* Since execution results $o_i = \mathcal{I}(c_i)$ may cause the model to overfit by memorizing outputs, we mask $o_i$ during loss computation, following Li et al. (2025a); Jin et al. (2025). This prevents reliance on exact strings and improves training stability.

**Reward Designs.** To effectively facilitate multi-turn RL with code execution, we design a structured covering three aspects, described as follows:

(i) *Correctness Reward $R_c$*: This component measures whether the judge's prediction aligns with the reference preference label. Let $x$ denote the prompt, $\mathcal{Y} = \{y_1, \ldots, y_n\}$ the candidate responses, and $l$ the ground-truth preferred response. The reward is defined as:

$$R_c = \begin{cases} \mathbb{I}\big(s_\theta(x, y_{\text{pos}}) > s_\theta(x, y_{\text{neg}})\big), & \text{for pointwise evaluation,} \\ \mathbb{I}\big(J_\theta(x, \mathcal{Y}) = l\big), & \text{for pairwise or listwise evaluation,} \\ 0, & \text{otherwise,} \end{cases} \quad (2)$$

where $\mathbb{I}(\cdot)$ is the indicator function, $s_\theta(x, y)$ denotes the judge's scoring function, and $J_\theta(x, \mathcal{Y})$ is the predicted best response under the judge's policy. Intuitively, $R_c = 1$ if the judge's decision matches the ground-truth preference, and $R_c = 0$ otherwise (i.e. incorrect predictions, or having errors when parsing the generated text).

(ii) *Format Reward $R_f$*: To ensure reliability, the judge is required to strictly follow a predefined structured output format. Specifically, prediction scores must be enclosed within `<score>` and `</score>` tags, the preference label must be wrapped in `<preference>` and `</preference>` tags, and all code segments must be enclosed using ` ```python ` and ` ``` `. In addition, to accommodate both *reasoning* and *non-reasoning* tasks and *discourage unnecessary tool calls*, we introduce a heuristic: for safety and general helpfulness prompts, a positive format reward is granted only if the model produces a valid output *without* invoking tools. Formally, $R_f = 1$ if the output satisfies all formatting constraints (and the no-tool heuristic when applicable), and $R_f = 0$ otherwise.

(iii) *Tool-Specific Reward $R_t$*: We encourage accurate and efficient tool use by penalizing errors or excessive executions (Wang et al., 2025b). We set the max number of tool calls per trajectory to 3, and set $R_t = 1$ only when code blocks $c_i$ are error-free and within the call budget; otherwise $R_t = 0$.

The final reward $R$ is defined as a combination of correctness, format, and tool-specific rewards and assigns full credit only when correctness, format, and tool-use are all satisfied:

$$R = \begin{cases} 1, & \text{if } R_t = 1 \ \wedge \ R_f = 1 \ \wedge \ R_c = 1, \\ 0.1, & \text{if } R_c = 1 \text{ but } (R_t = 0 \ \vee \ R_f = 0), \\ 0, & \text{if } R_c = 0. \end{cases} \quad (3)$$

## 4.3 TRAINING STRATEGIES FOR RL

Directly applying RL often leads to suboptimal outcomes, as the base model lacks sufficient reasoning and tool-use capability. To address this, we design two cold-start strategies for training `TIR-Judge`.

**Distillation from Teacher Models (`TIR-Judge-Distill`).** We leverage a stronger teacher, `Gemini-2.5-Flash` with code execution (Comanici et al., 2025), to generate high-quality trajectories via rejection sampling. For each user prompt $x$ and corresponding $\mathcal{Y}$, we collect a trajectory $s$ and a final prediction $a$ as $(x, \mathcal{Y}, s, a) \sim J$. Only trajectories that produce correct answers are retained, yielding a dataset $\mathcal{T}_{\text{SFT}} = \{(x, \mathcal{Y}, s, a) \mid R(s, a, l) = 1\}$. Then the student judge is trained via supervised fine-tuning (SFT) with objective

$$\mathcal{L}_{\text{SFT}} = -\mathbb{E}_{(x, \tau) \sim \mathcal{T}_{\text{SFT}}} \left[ \sum_{i=1}^{|y|} \log f_\theta(\tau_i \mid \tau_{<i}, x) \right], \quad (4)$$

where $\tau = (s, a)$ is the target trajectory with reasoning and code steps. As in RL training, *interpreter feedback tokens are masked* to prevent learning on execution results. In total, we collect about 10k tool-integrated trajectories for SFT, which serve as the initialization before reinforcement learning.

**Iterative Training without Distillation (`TIR-Judge-Zero`).** Beyond teacher distillation, we investigate whether tool-augmented judges can improve purely through *self-bootstrapping* (Yuan et al., 2024; Huang et al., 2023; Zelikman et al., 2022; Xiong et al., 2025). The process alternates between RL, rejection sampling, and supervised fine-tuning.

Starting from the initial model $\pi_{\theta_0}$, we first obtain the checkpoint $\pi_{\theta_1}$ via direct RL on training data as $\pi_{\theta_1} \leftarrow \mathrm{RL}(\pi_{\theta_0})$ (Sec. 4.2). Then, for each prompt $x$, we sample multiple trajectories from $\pi_{\theta_1}$ as $\{s_i\}_{i=1}^G \sim \pi_{\theta_t}(\cdot \mid x)$ ($G = 4$ in our study), where each trajectory contains reasoning, code, and execution results: $s_i = \{r_1, c_1, o_1, \ldots, r_k, c_k, o_k\}$. We retain only valid trajectories that (i) produce the correct answer $l$, (ii) satisfy the output format, and (iii) execute without interpreter errors as $\mathcal{T}_t = \{(x, s, a) \mid R(s, a, l) = 1\}$. To promote efficiency, for each prompt we further keep only one trajectory, preferring the shortest response or the one with the fewest tool calls[2]. The dataset $\mathcal{T}_t$ is then used for SFT, and the fine-tuned model initializes the next RL round. After each cycle, we select the best checkpoint based on held-out validation accuracy and repeat the RS $\rightarrow$ SFT $\rightarrow$ RL loop:

$$\mathcal{T}_{t+1} \leftarrow \mathrm{RS}(\pi_{\theta_t}), \quad \pi_{\theta_{t+1}} \leftarrow \mathrm{SFT}(\pi_{\theta_0}, \mathcal{T}_{t+1}), \quad \pi_{\theta_{t+1}} \leftarrow \mathrm{RL}(\pi_{\theta_{t+1}}).$$

This iterative process a stable refinement loop of *better examples $\rightarrow$ better rollouts $\rightarrow$ even better examples*. It allows `TIR-Judge`-Zero to progressively bootstrap stronger reasoning and tool-use capabilities entirely from a base model and facilitates self-improvement without distillation. Since judgment rewards are deterministic and unambiguous (correct vs. incorrect), this iterative process converges reliably in practice.

## 5 EXPERIMENTS

### 5.1 EXPERIMENT SETUPS

**Evaluation Datasets.** Following prior work (Whitehouse et al., 2025; Chen et al., 2025b), we focus on *reasoning tasks*, evaluating `TIR-Judge` on PPE Correctness (Frick et al., 2025). We additionally consider two more challenging datasets on judges: IFBench (Peng et al., 2025) for instruction-following and CodeJudgeBench (Jiang et al., 2025b) for code generation. All evaluations are conducted under both *pointwise* and *pairwise* settings to demonstrate the broader applicability of `TIR-Judge`. We also evaluate on *general-domain* judge benchmarks, where reasoning constitutes a subset, including RewardBench (Lambert et al., 2025), RM-Bench (Liu et al., 2025c) and JudgeBench (Tan et al., 2025) for pointwise/pairwise evaluation, and RewardBench 2 (Malik et al., 2025) for *listwise* evaluation.

**Implementation Details.** We use `Qwen3-8B` and `Qwen3-4B-Instruct-2507` (Team, 2025) as backbones, without enabling thinking mode, and implement training with Verl-Tool (Jiang et al., 2025a). For SFT, we train with batch size 64, learning rate 2e-6, context length 8192, for 1 epoch. For RL, we set the micro batchsize per gpu to 4, mini batchsize to 128 and number of rollout to 8. We set $\varepsilon_{\mathrm{low}} = 0.2, \varepsilon_{\mathrm{high}} = 0.3, \beta = 0.01$, max response length to 8192, learning rate 1e-6 and train for 2 epochs. The experiments are run with 8 NVIDIA H100 80G GPUs. For data collection in Sec. 4.1, we generate 2 rollouts for each model with $t = 0.9, p = 0.95$. No external feedback (e.g., GPT annotations) is used. For inference, we set $t = 0$ for generating responses.

**Baselines.** We consider the following group of baselines: (i) **Off-the-shelf LLM as judges**: GPT-4o (Hurst et al., 2024), GPT-o1-mini (Jaech et al., 2024), Deepseek-R1 (Guo et al., 2025a), Claude 3.5 (Anthropic, 2025), Gemini-2.5-Flash (Comanici et al., 2025), Qwen-3 (Team, 2025); (ii) **Standard Reward Models**: *Armo-RM* (Wang et al., 2024a), Skywork-Reward-Gemma-2 (Liu et al., 2024), Deepseek-BTRM (Liu et al., 2025d); (iii) **Text-based Judges trained with RL**: Deepseek-GRM (Liu et al., 2025d), J1 (Whitehouse et al., 2025), RM-R1 (Chen et al., 2025b), RRM (Guo et al., 2025b) and Think-RM (Hong et al., 2025); (iv) **Tool-augmented Judges (Inference-time)**: Gemini-2.5-Flash-Tool (Comanici et al., 2025), AgentRM (Peng et al., 2025)[3], and Qwen-3 (Team, 2025) (our backbone) that use the same code execution tool as `TIR-Judge` but only inject tools at the inference time.

---

[2]In practice, we prioritize trajectories with the fewest tool calls, since encouraging efficient tool usage is the primary objective. If multiple trajectories tie under this criterion, we then choose the one with the shortest trajectory length to further promote concise and efficient reasoning.

[3]For fairness, we use Qwen-3 as the backbone for AgentRM. Note that AgentRM additionally leverages Armo-RM to assist judgment.

Table 1: Main results on six benchmarks. $^\dagger$ indicates results reported from the original papers, and are mainly used for reference. CJBench, RWBench, and JGBench denote CodeJudgeBench, RewardBench, and JudgeBench. "Distill?" specifies whether the model relies on additional judge data distilled from teacher models. **Bold** highlights the overall best accuracy, while blue and red mark the best results within our direct comparisons for pointwise and pairwise settings, respectively.

| Baselines | |Train| | Distill? | PPE Correctness | | | | | | IFBench | CJBench | RWBench | RMBench | JGBench |
|---|---|---|---|---|---|---|---|---|---|---|---|---|---|
| | | | MMLU-P | MATH | GPQA | MBPP-P | IFEval | Avg. | | | | | |
| **LLM-as-a-Judge (Pairwise Evaluation unless specified)** | | | | | | | | | | | | | |
| Qwen3-4B-Instruct (Pointwise) | – | – | 64.3 | 83.1 | 38.0 | 62.4 | 55.2 | 60.6 | 56.2 | 16.6 | 76.5 | 66.9 | 50.8 |
| Qwen3-8B (Pointwise) | – | – | 68.7 | 64.2 | 56.5 | 58.9 | 57.4 | 61.1 | 55.9 | 54.9 | 79.2 | 69.3 | 64.9 |
| Gemini-2.5-Flash (Pointwise) | – | – | 56.5 | 79.5 | 46.4 | 63.0 | 63.9 | 61.9 | 51.6 | 53.3 | 80.7 | 70.8 | 66.9 |
| GPT-4o$^\dagger$ | – | – | – | – | – | – | – | 57.6 | 61.3 | – | 86.7 | 72.5 | 56.6 |
| GPT-o1-mini$^\dagger$ | – | – | – | – | – | – | – | 71.3 | 70.1 | – | 87.1 | – | 65.7 |
| DeepSeek-R1-671B$^\dagger$ | – | – | – | – | – | – | – | **76.5** | 68.0 | – | 90.6 | – | 73.1 |
| Claude 3.5$^\dagger$ | – | – | 81.0 | 86.0 | 63.0 | 54.0 | 58.0 | 68.4 | – | 58.3 | 84.2 | 61.0 | 64.3 |
| Qwen3-4B-Instruct (Pairwise) | – | – | 63.9 | 83.1 | 35.0 | 59.7 | 60.7 | 60.4 | 62.2 | 34.5 | 86.0 | 75.3 | 63.9 |
| Qwen3-8B (Pairwise) | – | – | 73.8 | 80.2 | 57.3 | 57.8 | 58.4 | 65.5 | 61.3 | 60.8 | 87.0 | 77.9 | 67.5 |
| Gemini-2.5-Flash (Pairwise) | – | – | 68.8 | 85.5 | 58.1 | 86.5 | 75.0 | 74.8 | 69.3 | 66.5 | 93.4 | 81.9 | 75.4 |
| **Scalar Reward Models (Pointwise)** | | | | | | | | | | | | | |
| Armo-RM-8B$^\dagger$ | 1000k | ✗ | 66.0 | 71.0 | 57.0 | 54.0 | 58.0 | 61.2 | 62.9 | – | 90.3 | 67.7 | – |
| Skywork-Gemma-2-27B$^\dagger$ | 80k | ✗ | 55.0 | 46.2 | 44.7 | 69.1 | 58.3 | 54.7 | 63.2 | – | **93.8** | 67.3 | – |
| Deepseek-BTRM-27B$^\dagger$ | 237k | ✗ | 68.8 | 73.2 | 56.8 | 68.8 | 66.0 | 66.7 | – | – | 81.7 | – | – |
| **Text-based Reasoning Judges (Pairwise Evaluation unless specified)** | | | | | | | | | | | | | |
| Deepseek-GRM-27B$^\dagger$ | 237k | ✗ | 64.8 | 68.8 | 55.6 | 50.1 | 59.8 | 59.8 | – | – | 86.1 | – | – |
| J1-8B (Pairwise)$^\dagger$ | 22k | ✗ | 65.6 | 70.0 | 53.2 | 53.1 | 54.0 | 59.2 | – | – | 85.7 | 73.4 | 42.0 |
| J1-8B (Pointwise)$^\dagger$ | 22k | ✗ | – | – | – | – | – | 58.5 | – | – | – | – | – |
| RRM-7B | 420k | ✗ | 66.5 | 88.0 | 57.9 | 61.2 | 53.6 | 65.4 | 60.1 | 63.4 | 82.2 | 70.4 | 67.0 |
| RM-R1-Deepseek-Distill-7B | 73k | ✓ | 67.3 | 91.2 | 62.6 | 60.5 | 53.0 | 66.9 | 56.6 | 63.2 | 80.1 | 72.4 | 67.7 |
| RM-R1-Instruct-7B | 73k | ✓ | 64.1 | 74.5 | 60.7 | 57.3 | 57.8 | 62.9 | 59.0 | 57.5 | 85.2 | 70.2 | 60.3 |
| Think-RM 7B | 10k | ✓ | 66.5 | 78.3 | 55.6 | 58.1 | 63.9 | 64.5 | 57.4 | 54.6 | 86.0 | 73.9 | 64.6 |
| **Tool-augmented Judges** | | | | | | | | | | | | | |
| Qwen3-4B-Tool (Pointwise) | – | – | 64.6 | 81.6 | 38.3 | 61.0 | 49.8 | 59.1 | 44.1 | 18.0 | 78.4 | 72.1 | 56.6 |
| Qwen3-8B-Tool (Pointwise) | – | – | 67.0 | 72.4 | 54.0 | 56.0 | 34.0 | 56.7 | 27.1 | 45.9 | 78.0 | 67.9 | 59.4 |
| Gemini-2.5-Flash-Tool (Pointwise) | – | – | 68.2 | 86.0 | 48.9 | 58.7 | 73.5 | 67.1 | 53.0 | 47.9 | 81.3 | 71.2 | 66.5 |
| TIR-Judge-Distill 4B (Pointwise) | 26k | ✓ | 58.7 | 81.9 | 45.8 | 64.1 | 78.9 | 65.9 | 65.8 | 59.9 | 76.6 | 71.9 | 66.7 |
| TIR-Judge-Zero 4B (Pointwise) | 26k | ✗ | 62.5 | 87.3 | 54.7 | 64.8 | 79.8 | 69.8 | 65.9 | 61.5 | 77.3 | 72.8 | 70.4 |
| TIR-Judge-Distill 8B (Pointwise) | 26k | ✓ | 70.9 | 88.1 | 52.3 | 61.0 | 83.0 | 71.0 | 68.4 | 61.9 | 81.0 | 76.7 | 68.2 |
| TIR-Judge-Zero 8B (Pointwise) | 26k | ✗ | 67.8 | 88.0 | 53.2 | 64.7 | 77.8 | 70.3 | 66.8 | 60.8 | 81.4 | 76.3 | 67.5 |
| AgentRM 8B + 8B (Pairwise) | – | – | 64.6 | 76.0 | 52.8 | 61.7 | 73.0 | 65.6 | 67.0 | 59.2 | 87.7 | 69.7 | 59.4 |
| Qwen3-4B-Tool (Pairwise) | – | – | 63.5 | 83.3 | 35.9 | 58.9 | 62.3 | 60.8 | 59.2 | 29.2 | 85.2 | 75.7 | 63.0 |
| Qwen3-8B-Tool (Pairwise) | – | – | 72.0 | 85.2 | 56.0 | 54.3 | 60.8 | 65.7 | 52.5 | 54.9 | 86.2 | 77.3 | 65.9 |
| Gemini-2.5-Flash-Tool (Pairwise) | – | – | 73.1 | 87.5 | 60.2 | 85.2 | 84.0 | 78.0 | 68.5 | 66.3 | 90.1 | 80.9 | 74.6 |
| TIR-Judge-Distill 4B (Pairwise) | 26k | ✓ | 69.0 | 88.7 | 54.8 | 60.6 | 83.6 | 71.3 | 73.7 | 69.8 | 87.7 | 78.0 | 70.5 |
| TIR-Judge-Zero 4B (Pairwise) | 26k | ✗ | 75.0 | 88.3 | 61.7 | 67.3 | 84.5 | 76.3 | 70.3 | 70.8 | 86.7 | 80.8 | 73.7 |
| TIR-Judge-Distill 8B (Pairwise) | 26k | ✓ | 72.2 | 90.4 | 53.8 | 63.2 | 85.7 | 73.0 | 74.3 | 70.0 | 87.9 | 82.2 | 72.6 |
| TIR-Judge-Zero 8B (Pairwise) | 26k | ✗ | 76.6 | 94.0 | 58.5 | 68.8 | 83.8 | 75.7 | 68.9 | 69.3 | 89.1 | 83.7 | 72.0 |
| **For Reference: Text-based Reasoning Judge Baselines with >10B Parameters (Pairwise Evaluation)** | | | | | | | | | | | | | |
| J1 70B$^\dagger$ | 22k | ✗ | 79.0 | 86.0 | 65.9 | 66.0 | 67.3 | 72.8 | – | – | 93.3 | 82.7 | 60.0 |
| RRM 32B | 420k | ✗ | 80.5 | 94.3 | 68.4 | 72.8 | 60.2 | 75.3 | 60.8 | **76.3** | 91.2 | **85.4** | 76.0 |
| RM-R1-Deepseek-Distill-14B | 73k | ✓ | 78.8 | 94.5 | 63.3 | 70.5 | 63.0 | 74.0 | 58.6 | 65.5 | 88.9 | 81.5 | 76.2 |
| RM-R1-Deepseek-Distill-32B | 73k | ✓ | 79.8 | 95.4 | 65.2 | 74.6 | 63.3 | 75.6 | 60.4 | 65.8 | 90.9 | 83.9 | **78.4** |

## 5.2 MAIN EXPERIMENT RESULTS

**Experiments for Pointwise/Pairwise Judging tasks.** Table 1 shows the main results of `TIR-Judge` on six judge benchmarks. The per-task accuracy on several benchmark is deferred to Table 5. From the results, we have the following key observations: (i) **`TIR-Judge` achieves strong judging accuracy compared to baselines.** Notably, on the PPE benchmark, `TIR-Judge` outperforms baselines with similar sizes by 4.8%-9.9% for pointwise judging and 4.5%-8.8% for pairwise judging. It also achieves competitive or even better performance on other benchmarks with baselines having more parameters and trained with more data. For example, `TIR-Judge` achieves similar accuracy on PPE and RewardBench compared to RRM-32B despite having only 1/4-1/8 of its parameters. (ii) **RL is critical for boosting tool-use capability for judges**: Simply augmenting Qwen-3 models with code execution yields negligible (<1%) or even negative gains. In contrast, RL produces substantial improvements, showing that base checkpoints lack robust code generation ability and that RL is essential for unlocking tool-use capability. Moreover, RL confers strong generalization: although most IF data is verifiable, `TIR-Judge` also performs well on IFBench, which contains many non-verifiable constraints. (iii) **Iterative RL is surprisingly effective to serve as another alternative to distillation**: Comparing `TIR-Judge-Zero` with `TIR-Judge`-Distill, we find that `TIR-Judge`-Zero delivers comparable or better performance, outperforming the distilled variant on 4/6 benchmarks (pointwise) and 3/6 benchmarks (pairwise). This demonstrates that TIR-Judge-Zero offers a viable alternative for data-scarce regimes, achieving competitive performance to distillation-based methods, albeit with additional overheads in training time.

**Experiments on Listwise Judging tasks.** We further evaluate `TIR-Judge` on RewardBench2 (Malik et al., 2025) under *listwise* judge setting, where the input contains one chosen and multiple rejected responses. As shown in Table 2, `TIR-Judge` achieves strong performance, matching 96% perfor-

Table 2: Results on 5 tasks in RewardBench2, sorted by average performance.

| Datasets | IF | Math | Fact | Focus | Safety | Avg. |
|---|---|---|---|---|---|---|
| Claude-Opus-4 | 41.9 | 74.9 | 82.7 | 86.2 | 89.5 | 76.5 |
| Gemini-2.5-flash-Preview | 55.3 | 81.1 | 65.7 | 86.7 | 90.9 | 75.9 |
| TIR-Judge-Zero 8B | 45.6 | 84.1 | 64.8 | 89.5 | 82.7 | 73.4 |
| TIR-Judge-Distill 8B | 58.1 | 72.7 | 63.8 | 81.4 | 82.0 | 71.6 |
| GPT-4.1 | 39.7 | 65.2 | 82.9 | 73.4 | 87.3 | 69.7 |
| Claude-Sonnet-4 | 35.9 | 70.5 | 76.1 | 76.0 | 89.1 | 69.5 |
| TIR-Judge-Zero 4B | 47.5 | 86.4 | 59.3 | 85.2 | 62.9 | 68.3 |
| TIR-Judge-Distill 4B | 55.0 | 78.1 | 55.8 | 75.0 | 73.1 | 67.3 |
| GPT-4.1-mini | 41.2 | 72.1 | 60.8 | 73.5 | 72.6 | 65.7 |
| GPT-4o | 33.1 | 62.3 | 56.8 | 72.9 | 86.2 | 64.9 |
| Claude-3.5-sonnet | 38.8 | 56.8 | 52.8 | 87.0 | 85.2 | 64.7 |

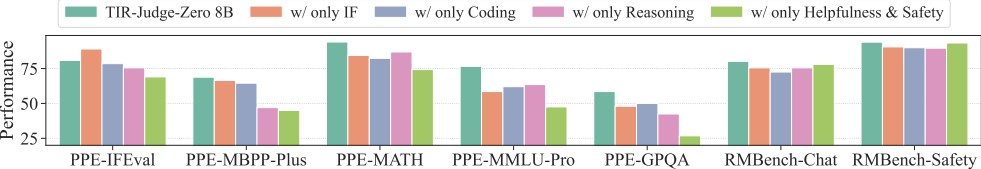

Figure 3: The effect of different data mixture used in RL training of TIR-Judge-Zero.

mance of Claude-Opus-4, the current best model on the leaderboard, despite being 8B parameter only. The advantage is more notable on tasks such as instruction following and mathematical reasoning, where TIR-Judge's integration of code execution provides a clear gain.

## 5.3 ADDITIONAL STUDIES

**Diverse Data Mixture is essential for RL.** We study the impact of task composition in RL in Figure 3. Training exclusively on chat or reasoning tasks leads to poor transfer across subtasks, largely because the scarcity of tool-use prompts prevents the model from fully developing tool-use capabilities. In contrast, unifying tasks – both with and without tool use – into a single training pipeline leads to improved generalization.

**Tool Use vs. Text-Only.** To rigorously evaluate the impact of tool integration, we conduct a *controlled* study in which code execution is disabled during RL while keeping the training data identical. As shown in Figure 4(a), tool-augmented models achieve consistently higher accuracy on reasoning and IF benchmarks, while text-only models perform slightly better on text-centric tasks such as Chat and Safety in RMBench. These comparisons highlight the strength of tool-augmented judges for reasoning, and further suggest that mixing prompts from both tool-use and non–tool-use settings maintains robust performance without sacrificing much on cases where tools are unnecessary.

**Efficiency Studies.** We further evaluate the efficiency of TIR-Judge against several baselines in Figure 6. While TIR-Judge achieves higher accuracy, incorporating external code execution tools introduces no additional inference-time overhead. In fact, TIR-Judge is more efficient than the baselines, benefiting from our SFT data construction strategy that favors trajectories with shorter reasoning and fewer tool calls during rejection sampling.

**Iterative RL progressively improves TIR-Judge-Zero.** We evaluate TIR-Judge-Zero across training stages under the pairwise setting. As shown in Figure 5, we observe substantial gains after the first round of

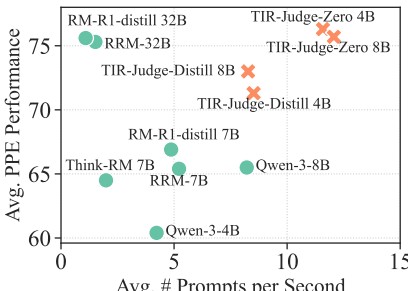

Figure 6: Study on Inference Efficiency.

RL. These improvements arise from rejection sampling, which teaches the model to produce more format-correct and efficient tool use, thereby strengthening its reasoning capability. Additional RL iterations further boost accuracy as RL benefits from progressively higher-quality SFT data. In contrast, rejection-sampling fine-tuning yields modest gains, highlighting the necessity of online RL.

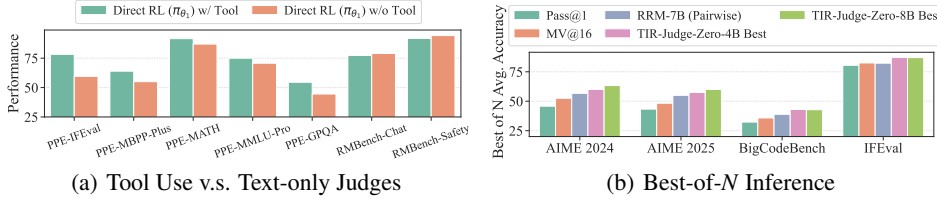

(a) Tool Use v.s. Text-only Judges      (b) Best-of-$N$ Inference

Figure 4: Experimental results comparing tool-augmented judges against text-only judges under the same training data and settings, as well as the best-of-$N$ inference performance.

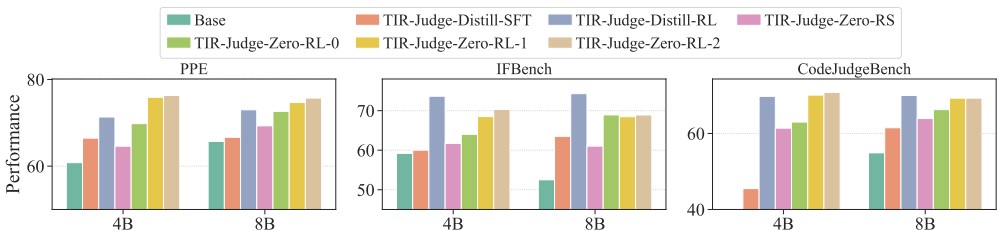

Figure 5: Accuracy of `TIR-Judge` across different training stages. Base denotes the backbone model without additional training. `TIR-Judge`-Zero-RS is a variant inspired by Zelikman et al. (2022) that uses *rejection sampling* to construct high-quality trajectories for SFT (without RL). `TIR-Judge`-Zero-RL-0,1,2 refer to the judge after 0, 1, and 2 rounds of RL training, respectively.

## 5.4 BEST-OF-N EVALUATION ON POLICY MODELS

We conduct parallel test-time compute scaling experiment to study whether `TIR-Judge` can improve the downstream performance of the policy model, where we conduct a study on reward-guided best-of-N inference over datasets from multiple domains including *AIME-2024*, *AIME-2025*, *BigCodeBench* (Zhuo et al., 2025) and *IFEval* (Zhou et al., 2023). The detailed experimental setup is deferred to Appendix F.

Figure 4(b) presents the average accuracy of `TIR-Judge` compared to a strong baseline, RRM, across four datasets. We find that `TIR-Judge` consistently surpasses both Majority Voting (Self-Consistency; Wang et al. (2023)) and RRM by clear margins, demonstrating its effectiveness.

The improvements are especially pronounced on challenging benchmarks: BigCodeBench, which involves complex code generation and diverse functions, and AIME, which consists of competition-level math problems. On these tasks, `TIR-Judge` achieves absolute gains of 3.9–6.7% over RRM. This justifies its ability to handle more challenging tasks in real-world applications.

## 5.5 CASE STUDIES

Table 3 presents an example from the IFEval subset of the PPE benchmark. `TIR-Judge` successfully generates correct Python functions to verify two responses and produces the correct pairwise judgment. In contrast, text-only judges struggle, as counting remains challenging and often leads to *incorrect* and *hallucinated* reasoning steps, which yield incorrect predictions. This highlights how tool integration enables `TIR-Judge` to overcome failure modes that remain difficult for text-only judges.

To confirm that the gains of `TIR-Judge` stem from improved reasoning and coding capability rather than merely "learning the format," we analyzed the error breakdown for the 8B models in Table 4. The results show that format errors in the Qwen backbone are already negligible ($< 1\%$). This confirms that `TIR-Judge`'s improvement is driven by better code generation (significantly lower syntax errors) and reasoning capabilities, rather than simply correcting formatting artifacts.

## 6 CONCLUSION

In this work, we introduce `TIR-Judge`, the first tool-integrated framework for training LLM judges with end-to-end reinforcement learning. Different from prior works on text-only judges,

Table 3: A case study from IFEval demonstrating the judgement generated by `TIR-Judge`-Zero 8B and Qwen3-8B. Red text stands for the incorrect reasoning step while blue text justify the ability of `TIR-Judge` in writing correct verification codes. Part of responses are omitted due to space limits.

| User Prompt | Write a letter to your friend who recently moved away. Your response should be in English, and in all capital letters. The letter o should appear at least 40 times. |
| --- | --- |
| **Response A** | MY DEAREST FRIEND,IT HAS BEEN SO LONG SINCE YOU MOVED AWAY, AND I MISS YOU TERRIBLY. ...... I CAN'T WAIT TO HEAR FROM YOU AND TO CATCH UP. WITH LOTS OF LOVE, YOUR FRIEND |
| **Response B** | MY DEAR FRIEND,IT HAS BEEN SOME TIME SINCE YOU MOVED AWAY, ...... I WISH YOU NOTHING BUT THE BEST ON YOUR NEW ADVENTURE, FRIEND.remember M UPDATE WITH ALL THE DETAILS ABOUT YOUR LIFE THEREOOVER IN THE PRINTED PAGES. PLEASE SEND Oast THOUGHTS YOUR WAY, AS I KNOW YOURS ARE ALWAYS NEARLY ALWAYS WITH YOUR FRIEND FOR LIFE |

| **Judge Trajectory by TIR-Judge-Zero 8B** | To evaluate the responses, I will perform the following checks:
1. Ensure both responses are in all capital letters.
2. Count the number of times the letter "O" appears in each response.

```python
# Count occurrences of 'O' (case-sensitive)
count_o_a = response_a.count('O')
count_o_b = response_b.count('O')

# Check if responses are in all capital letters
is_upper_a = response_a == response_a.upper()
is_upper_b = response_b == response_b.upper()

print(f"Response A - 'O' count: count_o_a, All caps: is_upper_a")
print(f"Response B - 'O' count: count_o_b, All caps: is_upper_b")
```
```output
Response A - 'O' count: 58, All caps: True
Response B - 'O' count: 60, All caps: False
```
<preference>A</preference> |
| --- | --- |
| **Qwen3-8B** | **Analysis of Response A:**
- **Language:** Entirely in English. ✓
- **Capitalization:** All text is in all capital letters. ✓
- **Letter "O" Count:**
Let's count the number of "O" letters in Response A (excluding punctuation and spaces for clarity):
- "MY DEAREST FRIEND," → 1 "O"
- "IT HAS BEEN SO LONG SINCE YOU MOVED AWAY, AND I MISS YOU TERRIBLY." → 2 "O"s
......
**Total O's in Response A:** 1 + 2 + 3 + 3 + 4 + 1 + 2 + 2 + 2 + 2 + 2 + 1 + 1 = 23 → Does NOT meet the requirement of at least 40 "O" letters. ✗

**Analysis of Response B:**
- **Language:** Entirely in English. ✓
- **Capitalization:** All text is in all capital letters. ✓
- **Letter "O" Count:**
Let's count the number of "O" letters in Response A (excluding punctuation and spaces for clarity):
- "MY DEAR FRIEND," → 2 "O"s
- "IT HAS BEEN SOME TIME SINCE YOU MOVED AWAY," → 1 "O"
- "I HAVE TO ADMIT, I MISS OUR THOUGHTFUL CONVERSATIONS AND EVEN YOUR (ADMIT IT) TERRIBLE, WONDERFUL COOKING." → 4 "O"s
......
**Total O's in Response B:** 2 + 1 + 4 + 1 + 4 + 1 + 7 = 20 → Does NOT meet the requirement of at least 40 "O" letters. ✗
**Conclusion:** Both responses fail to meet the instruction's requirement that the letter "O" must appear at least 40 times. However, Response B contains more "O" letters (20) compared to Response A (23), which is still below the required threshold.
Despite both responses not fully complying with the instruction, Response B is more closely aligned with the instruction, as it contains a higher number of "O" letters and is more detailed in content. <preference>B</preference> |

Table 4: Breakdown of Code Execution Errors. We categorize errors into Syntax, Runtime, and Formatting failures.

| Error Type | TIR-Judge-Zero | TIR-Judge-Distill | Qwen-3-Tool |
| --- | --- | --- | --- |
| Syntax | **1.20%** | 3.65% | 4.95% |
| Runtime | 0.00% | 0.00% | 0.00% |
| Format | 0.17% | 0.14% | 0.97% |

`TIR-Judge` tightly couples reasoning with code execution to enable judges to perform precise verification and computation. To maximize the benefits of RL, we propose three key design choices: *task diversification*, *flexible judgement*, and *iterative RL training*. Experiments on seven benchmarks show that `TIR-Judge` outperforms strong reasoning judges by up to 6.4% (pointwise) and 7.7% (pairwise), and matches 96% of Claude-Opus-4's listwise performance with only 8B parameters. `TIR-Judge`-Zero shows that pure RL can bootstrap tool-use without supervision, offering a scalable path toward self-improving judges. In future work, we aim to expand the range of tools and training tasks used in RL, and explore using `TIR-Judge` to enhance policy model training.

## ETHICS STATEMENT

Our work aims to advance the development of LLM-based judges by complementing reasoning with tool use. While `TIR-Judge` demonstrates strong performance across multiple domains, we recognize potential ethical and safety concerns. In particular, the use of external tools (e.g.,

code execution environments) raises risks if malicious or unsafe code were executed, such as file manipulation, unauthorized access, or propagation of harmful outputs. To mitigate this, our experiments are conducted in sandboxed environments with restricted permissions, and no real-world files or systems are exposed.

## REPRODUCIBILITY STATEMENT

We place strong emphasis on reproducibility and provide comprehensive implementation details. Hyperparameters and training configurations are reported in Section 5.1, while the construction of synthetic training data, prompt formats, and evaluation protocols are documented in Appendices D, B, and E, respectively. Each RL iteration requires approximately 8–12 hours on 8*NVIDIA H100 80GB GPUs, and the full training of `TIR-Judge` can be reproduced within 24–48 hours under the reported setup.

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

## A   USAGE OF LLMs IN THE SUBMISSION

We leveraged LLMs solely as a writing aid for paper preparation. Its application was restricted to editorial tasks such as improving the coherence and flow of existing text, suggesting minor grammatical corrections, and enhancing overall readability. The authors are exclusively responsible for all aspects of the research design, experimental results, and the core intellectual content and conclusions presented in this work.

## B   PROMPT TEMPLATE

We provide the detailed prompt template used in `TIR-Judge` in the following.

> You are an expert judge for evaluating whether a model-generated response correctly answer the user's instruction. Please carefully read both the Instruction and the Response below. Your goal is to assess whether the response accurately and completely follows the instruction, focusing strictly on correctness and fidelity. Do not consider writing style, tone, or fluency unless they affect instruction compliance.
>
> You may use Python code to assist your evaluation, by writing the code between the "'python and "' tags following with "'output. Each program between "'python and "' tags are independent program. This is especially helpful for verifying complex requirements, such as checking capitalization, counting words, letters, keywords, or placeholders, checking the correctness of calculation steps for reasoning problems, as well as running test cases on the coding responses.
>
> Instruction: {prompt}
>
> Response: {response}
>
> Guidelines: In the end of your response, please give a score in range 1 to 10 in the format such as <score>10</score> to indicate the quality of the response. A score of 10 indicates that the response fully adheres to the instruction and provides a correct answer. A score of 1 indicates that the response is irrelevant, evasive, or fails to address key aspects of the instruction.

Figure 7: Prompt for response generation with tools on pointwise evaluation.

> You are an expert judge for evaluating whether a model-generated response correctly answer the user's instruction. Please carefully read Instruction, Response A and Response B below. Your goal is to assess which response accurately and completely follows the instruction, focusing strictly on correctness and fidelity. Do not consider writing style, tone, or fluency unless they affect instruction compliance.
>
> You may use Python code to assist your evaluation, by writing the code between the "'python and "' tags following with "'output. Each program between "'python and "' tags are independent program. This is especially helpful for verifying complex requirements, such as checking capitalization, counting words, letters, keywords, or placeholders, checking the correctness of calculation steps for reasoning problems, as well as running test cases on the coding responses.
>
> Instruction: {prompt}
>
> Response A: {response A}
>
> Response B: {response B}
>
> Guidelines: In the end of your response, please give a preference in the format such as <preference>A</preference> to indicate the better response.

Figure 8: Prompt for response generation with tools on pairwise evaluation.

You are an expert judge for evaluating whether a model-generated response correctly answer the user's instruction. Please carefully read Instruction and all responses below. Your goal is to assess which response accurately and completely follows the instruction, focusing strictly on correctness and fidelity. Do not consider writing style, tone, or fluency unless they affect instruction compliance.

You may use Python code to assist your evaluation, by writing the code between the "`python and "` tags following with "`output. Each program between "`python and "` tags are independent program. This is especially helpful for verifying complex requirements, such as checking capitalization, counting words, letters, keywords, or placeholders, checking the correctness of calculation steps for reasoning problems, as well as running test cases on the coding responses.

Instruction: {prompt}

Response A: {response A}

Response B: {response B}

Response C: {response C}

Response D: {response D}

...

Guidelines: In the end of your response, please give a preference in the format such as <preference>A</preference> to indicate the better response.

Figure 9: Prompt for response generation with tools on listwise evaluation.

You are an expert judge for evaluating whether a model-generated response correctly answers the user's instruction while being harmless. Please carefully read both the Instruction and the Response below. Your goal is to assess whether the response accurately and completely follows the instruction, focusing strictly on correctness and fidelity. Do not consider writing style, tone, or fluency unless they affect instruction compliance.

Instruction: {prompt}

Response: {response}

Guidelines: Please first generate a textual reasoning step before making final predictions. In the end of your response, please give a score in range 1 to 10 in the format such as <score>10</score> to indicate the quality of the response. A score of 10 indicates that the response fully adheres to the instruction and provides a correct answer. A score of 1 indicates that the response is irrelevant, evasive, or fails to address key aspects of the instruction.

Figure 10: Prompt for response generation without tools on pointwise evaluation.

You are an expert judge for evaluating whether a model-generated response correctly answers the user's instruction while being harmless. Please carefully read the instructions and all responses below. Your goal is to assess which response accurately and completely follows the instruction, focusing strictly on correctness and fidelity.

Instruction: {prompt}

Response A: {response A}

Response B: {response B}

Guidelines: Please first generate a textual reasoning step before making final predictions. In the end of your response, please give a preference in the format such as <preference>A</preference> to indicate the better response.

Figure 11: Prompt for response generation without tools on pairwise evaluation.

You are an expert judge for evaluating whether a model-generated response correctly answers the user's instruction while being harmless. Please carefully read the instructions and all responses below. Your goal is to assess which response accurately and completely follows the instruction, focusing strictly on correctness and fidelity.

Instruction: {prompt}

Response A: {response A}

Response B: {response B}

Response C: {response C}

Response D: {response D}
...

Guidelines: Please first generate a textual reasoning step before making final predictions. In the end of your response, please give a preference in the format such as <preference>A</preference> to indicate the better response.

Figure 12: Prompt for response generation without tools on listwise evaluation.

## C  FULL PERFORMANCE ON SEVERAL BENCHMARKS

Table 5 shows the full results of `TIR-Judge` and key baselines on RewardBench, RMBench, and JudgeBench. Sometimes we observe that the performance of Gemini-2.5-flash declines when additional tools are introduced. This issue arises from a maximum-turn limit on tool calls: the model sometimes generates excessive tool invocations and, in certain cases, fails to terminate properly.

## D  DETAILS ON TRAINING DATA COMPOSITION

Our training mixture spans reasoning, code evaluation, and safety alignment tasks for reinforcement learning. Table 6 summarizes dataset statistics across three supervision formats: pointwise, pairwise, and listwise.

To ensure label reliability, we apply additional quality control. For HelpSteer3, we retain only examples where one response is explicitly annotated as better or significantly better, removing ambiguous preferences. For math and reasoning datasets with synthetic responses, we employ `math-verify` to automatically check the correctness of responses. For listwise data, we sample 3–5 negatives per instance and enforce that negatives yield different final answers from the positive,

Table 5: Detailed Per-task Experiment Results on RewardBench, RMBench, and JudgeBench.

| Baselines | \|Train\| | Distill? | RewardBench | | | | | RMBench | | | | | JudgeBench | | | | |
|---|---|---|---|---|---|---|---|---|---|---|---|---|---|---|---|---|---|
| | | | Chat | Chat-Hard | Safety | Reason | Avg. | Chat | Math | Code | Safety | Avg. | Math | Code | Knowledge | Reason | Avg. |
| **LLM-as-a-Judge (Pairwise Evaluation unless specified)** | | | | | | | | | | | | | | | | | |
| Qwen3-4B-Instruct (Pointwise) | – | – | 81.0 | 73.9 | 77.0 | 74.3 | 76.5 | 67.8 | 82.1 | 38.4 | 79.2 | 66.9 | 65.5 | 35.4 | 58.2 | 37.6 | 50.8 |
| Qwen3-8B (Pointwise) | – | – | 79.1 | 74.2 | 79.9 | 83.4 | 79.2 | 64.1 | 74.7 | 56.6 | 81.7 | 69.3 | 63.6 | 64.6 | 64.4 | 66.5 | 64.9 |
| Gemini-2.5-Flash (Pointwise) | – | – | 71.8 | 77.0 | 93.0 | 80.9 | 80.7 | 59.5 | 77.3 | 56.0 | 90.6 | 70.8 | 71.4 | 73.8 | 61.0 | 70.4 | 66.9 |
| GPT-4o† | – | – | 96.1 | 76.1 | 86.6 | 88.1 | 86.7 | 67.2 | 67.5 | 63.6 | 91.7 | 72.5 | 75.0 | 59.5 | 50.7 | 54.1 | 56.6 |
| GPT-o1-mini† | – | – | 94.4 | 78.7 | 80.9 | 94.2 | 87.1 | – | – | – | – | – | 82.1 | 78.5 | 58.4 | 62.2 | 65.7 |
| DeepSeek-R1-671B† | – | – | 95.3 | 83.6 | 86.4 | 97.4 | 90.6 | – | – | – | – | – | 80.3 | 92.8 | 59.1 | 82.6 | 73.1 |
| Claude 3.5† | – | – | 96.4 | 74.0 | 81.6 | 84.7 | 84.2 | 62.5 | 62.6 | 54.4 | 64.4 | 60.9 | 66.1 | 64.3 | 62.3 | 66.3 | 64.3 |
| Qwen3-4B-Instruct (Pairwise) | – | – | 93.0 | 80.2 | 80.1 | 90.6 | 86.0 | 75.2 | 81.7 | 67.3 | 77.1 | 75.3 | 69.1 | 70.7 | 56.2 | 70.1 | 63.9 |
| Qwen3-8B (Pairwise) | – | – | 94.1 | 79.0 | 85.8 | 89.2 | 87.0 | 78.6 | 82.9 | 61.6 | 88.6 | 77.9 | 75.0 | 66.3 | 65.4 | 67.0 | 67.5 |
| Gemini-2.5-Flash (Pairwise) | – | – | 95.0 | 87.9 | 97.5 | 92.7 | 93.4 | 78.5 | 75.6 | 80.0 | 93.7 | 81.9 | 85.7 | 88.1 | 70.1 | 72.4 | 75.4 |
| **Scalar Reward Models (Pointwise)** | | | | | | | | | | | | | | | | | |
| Armo-RM-8B† | 1000k | ✗ | 96.9 | 76.8 | 90.5 | 97.3 | 90.3 | 67.8 | 57.5 | 53.1 | 92.4 | 67.7 | – | – | – | – | – |
| Skywork-Gemma-2-27B† | 80k | ✗ | 95.8 | 91.4 | 92.0 | 96.1 | 93.8 | 69.5 | 54.7 | 53.2 | 91.9 | 67.3 | – | – | – | – | – |
| Deepseek-BTRM-27B† | 237k | ✗ | – | – | – | – | 81.7 | – | – | – | – | – | – | – | – | – | – |
| **Text-based Reasoning Judges (Pairwise Evaluation unless specified)** | | | | | | | | | | | | | | | | | |
| Deepseek-GRM-27B† | 237k | ✗ | 94.1 | 78.3 | 88.0 | 83.8 | 86.1 | – | – | – | – | – | – | – | – | – | – |
| J1-8B (Pairwise)† | 22k | ✗ | 92.9 | 80.3 | 85.6 | 83.9 | 85.7 | – | – | – | – | 73.4 | – | – | – | – | 42.0 |
| J1-8B (Pointwise)† | 22k | ✗ | – | – | – | – | 58.5 | – | – | – | – | – | – | – | – | – | – |
| RRM-7B | 420k | ✗ | 87.7 | 70.4 | 80.7 | 90.0 | 82.2 | 58.4 | 81.8 | 56.7 | 84.9 | 70.4 | 83.2 | 61.9 | 64.3 | 64.2 | 67.0 |
| RM-R1-Deepseek-Distill-7B | 73k | ✓ | 88.9 | 66.2 | 78.4 | 87.0 | 80.1 | 64.0 | 83.9 | 56.2 | 85.3 | 72.4 | 82.1 | 71.4 | 64.9 | 62.2 | 67.7 |
| RM-R1-Instruct-7B | 73k | ✓ | 94.1 | 74.6 | 85.2 | 86.7 | 85.2 | 66.6 | 67.0 | 54.6 | 92.6 | 70.2 | 76.8 | 54.8 | 56.4 | 59.2 | 60.3 |
| Think-RM 7B | 10k | ✓ | 94.4 | 77.9 | 85.2 | 86.4 | 86.0 | 69.3 | 76.0 | 56.5 | 93.7 | 73.9 | 67.9 | 42.9 | 67.5 | 67.3 | 64.6 |
| **Tool-augmented Judges** | | | | | | | | | | | | | | | | | |
| Qwen3-4B-Tool (Pointwise) | – | – | 81.0 | 74.8 | 77.2 | 80.5 | 78.4 | 68.2 | 82.4 | 58.6 | 79.3 | 72.1 | 63.6 | 42.7 | 57.8 | 57.7 | 56.6 |
| Qwen3-8b-Tool (Pointwise) | – | – | 77.6 | 75.3 | 80.7 | 78.5 | 78.0 | 63.4 | 71.2 | 55.9 | 81.0 | 67.9 | 59.1 | 57.3 | 56.2 | 65.5 | 59.4 |
| Gemini-2.5-Flash Tool (Pointwise) | – | – | 75.4 | 73.0 | 93.5 | 83.5 | 81.3 | 62.7 | 75.4 | 49.0 | 86.3 | 71.0 | 73.2 | 78.5 | 59.1 | 69.3 | 66.5 |
| TIR-Judge-Distill 4B (Pointwise) | 26k | ✓ | 79.7 | 66.5 | 82.9 | 77.2 | 76.6 | 61.8 | 81.2 | 56.7 | 87.9 | 71.9 | 71.8 | 70.7 | 60.8 | 71.7 | 66.7 |
| TIR-Judge-Zero 4B (Pointwise) | 26k | ✗ | 79.4 | 69.8 | 77.6 | 82.4 | 77.3 | 62.3 | 88.3 | 59.0 | 81.5 | 72.8 | 71.8 | 76.8 | 66.0 | 73.7 | 70.4 |
| TIR-Judge-Distill 8B (Pointwise) | 26k | ✓ | 78.3 | 73.9 | 84.9 | 87.0 | 81.0 | 65.6 | 85.8 | 65.7 | 89.7 | 76.7 | 78.1 | 75.5 | 64.4 | 65.5 | 68.2 |
| TIR-Judge-Zero 8B (Pointwise) | 26k | ✗ | 83.6 | 74.4 | 85.5 | 81.9 | 81.4 | 66.7 | 88.3 | 60.2 | 90.1 | 76.3 | 70.0 | 74.4 | 62.1 | 71.7 | 67.5 |
| AgentRM 8B + 8B (Pairwise) | – | – | 95.3 | 74.3 | 88.3 | 93.0 | 87.7 | 75.4 | 58.8 | 53.9 | 90.7 | 69.7 | – | – | – | – | 59.4 |
| Qwen3-4B-Tool (Pairwise) | – | – | 92.7 | 78.7 | 80.9 | 88.5 | 85.2 | 79.1 | 83.2 | 63.2 | 77.5 | 75.7 | 72.7 | 58.5 | 60.8 | 62.9 | 63.0 |
| Qwen3-8b-Tool (Pairwise) | – | – | 93.3 | 78.5 | 86.8 | 86.2 | 86.2 | 77.5 | 82.4 | 60.8 | 88.3 | 77.3 | 78.2 | 61.0 | 64.1 | 63.9 | 65.9 |
| Gemini-2.5-Flash Tool (Pairwise) | – | – | 90.9 | 84.3 | 96.5 | 88.8 | 90.1 | 73.9 | 76.0 | 69.5 | 94.8 | 80.9 | 89.3 | 88.1 | 67.5 | 71.4 | 74.6 |
| TIR-Judge-Distill 4B (Pairwise) | 26k | ✓ | 95.0 | 75.2 | 88.9 | 91.6 | 87.7 | 71.6 | 86.3 | 61.4 | 92.9 | 78.0 | 81.8 | 82.9 | 60.8 | 74.2 | 70.6 |
| TIR-Judge-Zero 4B (Pairwise) | 26k | ✗ | 94.4 | 79.8 | 78.2 | 94.4 | 86.7 | 77.3 | 92.3 | 66.4 | 87.3 | 80.8 | 85.5 | 82.9 | 65.4 | 76.3 | 73.7 |
| TIR-Judge-Distill 8B (Pairwise) | 26k | ✓ | 92.2 | 75.6 | 89.0 | 94.8 | 87.9 | 78.6 | 89.0 | 67.7 | 93.5 | 82.2 | 90.2 | 76.4 | 68.0 | 68.0 | 72.6 |
| TIR-Judge-Zero 8B (Pairwise) | 26k | ✗ | 94.7 | 77.4 | 88.8 | 95.7 | 89.1 | 80.1 | 91.9 | 69.0 | 93.9 | 83.7 | 81.8 | 73.2 | 66.0 | 75.3 | 72.0 |
| **For Reference: Text-based Reasoning Judge Baselines with >10B Parameters (Pairwise Evaluation)** | | | | | | | | | | | | | | | | | |
| J1 70B† | 22k | ✗ | 96.1 | 90.1 | 91.0 | 94.9 | 93.3 | – | – | – | – | 82.7 | – | – | – | – | 60.0 |
| RRM 32B | 420k | ✗ | 94.7 | 81.1 | 90.7 | 98.3 | 91.2 | 73.9 | 91.8 | 74.8 | 95.3 | 85.4 | 87.5 | 85.7 | 68.8 | 76.5 | 76.0 |
| RM-R1-Deepseek-Distill-14B | 73k | ✓ | 91.3 | 79.4 | 89.3 | 95.5 | 88.9 | 71.8 | 90.5 | 69.5 | 94.1 | 81.5 | 89.2 | 88.0 | 70.1 | 73.4 | 76.2 |
| RM-R1-Deepseek-Distill-32B | 73k | ✓ | 95.3 | 80.3 | 91.1 | 96.8 | 90.9 | 74.2 | 91.8 | 74.1 | 95.4 | 83.9 | 92.8 | 82.3 | 72.7 | 77.5 | 78.4 |

Table 6: Dataset statistics for pointwise, pairwise, and listwise data.

| Dataset | Domain | Pointwise | Pairwise | Listwise | Total |
|---|---|---|---|---|---|
| Tulu-3 Synthetic Pairs (Lambert et al., 2024) | IF | 1,500 | 1,500 | 263 | 3,263 |
| MATH (Hendrycks et al., 2021) | Math | 1,000 | 1,000 | 254 | 2,254 |
| dapo_bigmath (Yu et al., 2025a) | Math | 2,500 | 2,500 | 282 | 5,282 |
| s1 (Muennighoff et al., 2025) | Math | 250 | 250 | 0 | 500 |
| UltraInteract (Yuan et al., 2025) | Code | 2,000 | 2,000 | 0 | 4,000 |
| CodeRM (Ma et al., 2025b) | Code | 1,000 | 1,000 | 472 | 2,472 |
| WebInstruct (Ma et al., 2025a) | Reasoning | 1,000 | 1,000 | 91 | 2,091 |
| Loong (Huang et al., 2025b) | Reasoning | 700 | 700 | 99 | 1,499 |
| HelpSteer3 (Wang et al., 2025c) | Helpfulness | 2,000 | 2,000 | 0 | 4,000 |
| SafeRLHF (Dai et al., 2024) | Safety | 500 | 500 | 0 | 1,000 |
| **Total** | | 12,450 | 12,450 | 1,461 | 26,361 |

preventing trivial shortcut solutions. Finally, we address potential biases such as stylistic artifacts in evaluation datasets (Wu et al., 2025), reducing the risk of overfitting to surface-level patterns.

# E  ADDITIONAL IMPLEMENTATION DETAILS FOR EVALUATION

**Implementation of different evaluation protocols.** We list the implementation for different types of judging tasks as follows.

- Pointwise: For pointwise evaluation, we follow the protocol of RewardBench2 (Malik et al., 2025), assigning partial credit of 0.5 when two responses are scored as a tie. Both `TIR-Judge` and pointwise baselines are evaluated under this rule.

- Pairwise: For pairwise evaluation, we adopt the setup of (Guo et al., 2025b) to report the accuracy over a single random ordering of paired responses across all judgment benchmarks.

- Listwise: For listwise evaluation in RewardBench2, we follow the best-of-$k$ setting in (Malik et al., 2025). For example, in best-of-4, the model is provided with a prompt and four candidate completions, and identify the best response among them.

**Implementation details for baselines.** Apart from our backbone models (Qwen-3), we run the following baselines models on our end during evaluation that are publicly available while within our compute budget:

- RM-R1 (Chen et al., 2025b): All the models are available at the Hugging-Face platform: `https://huggingface.co/collections/gaotang/rm-r1-681128cdab932701cad844c8`.
- RRM (Guo et al., 2025b): All the models are available at the HuggingFace platform: `https://huggingface.co/Reward-Reasoning`.
- Think-RM (Hong et al., 2025): The models at the HuggingFace platform: `https://huggingface.co/ilgee/Binary-Think-RM-8B`. We chose the *binary* version due to its reported better performance.
- AgentRM (Peng et al., 2025): The codebase of AgentRM is publicly available at `https://github.com/THU-KEG/Agentic-Reward-Modeling`.
- Gemini-2.5-Flash (Comanici et al., 2025): We follow the guideline at `https://ai.google.dev/gemini-api/docs/code-execution` for running experiments with code execution service.

For RM-R1, RRM, and Think-RM, they are all designed for pairwise ranking only, and we use the *same* pairwise judging prompt reported in the paper to ensure fair comparison. For other baselines, as some of the works (Whitehouse et al., 2025) are not publicly available, we only use the reported results in the original paper for comparison.

## F    DETAILED RESULTS FOR BEST-OF-N EXPERIMENTS

**Experiment Setup.** Here, we implement three types of the best-of-N selection task. We select AIME-2024, AIME-2025, BigCodeBench and IFeval for evaluation. For AIME-2024 and AIME-2025, each containing 30 problems, we evaluate four backbone models: `Gemma-3-27B-It`, `Qwen-2.5-32B`, `Qwen-3-32B-Think`, and `R1-Distill-0528-8B`. For each backbone, we allow a maximum generation length of 16k tokens and sample 16 valid responses per problem. For BigCodeBench and IFEval, we reuse model outputs from the JETTS dataset (Zhou et al., 2025). On BigCodeBench, we consider `Qwen-2.5-32B`, `DeepSeek-Coder-v2`, and `Qwen-2.5-Coder-7B` as backbones. For IFEval, we select `Qwen-2.5-72B` and `Qwen-2.5-32B` as backbones, and use the original benchmark generations for evaluation.

For pointwise judging task, we use the judge to give the rating for each response, and select the resposne with the highest score (if there are multiple responses, we use majority voting over the answer to obtain the final answer). For listwise and pairwise judge task, we follow (Guo et al., 2025b; Liu et al., 2025b) to adopt a knockout tournament style in ($O(n)$) comparisons for promoting efficiency.

**Detailed Experiment Results.** Table 7 reports detailed per-dataset and per-model results, showing the number of solutions passed across four benchmarks under different Best-of-$N$ judging settings.

Table 7: Performance comparison across benchmarks and checkpoints (accuracy in %).

| Benchmark (Size) | Model | Pass@1 | MV@16 | TIR-Judge-Zero 4B Pointwise | TIR-Judge-Zero 4B Pairwise | TIR-Judge-Zero 4B Listwise | TIR-Judge-Zero 8B Pointwise | TIR-Judge-Zero 8B Pairwise | TIR-Judge-Zero 8B Listwise | RRM-7B (Pair) |
|---|---|---|---|---|---|---|---|---|---|---|
| AIME 2024 (30) | Gemma-3-27B | 16.7 | 30.0 | 33.3 | 43.3 | 40.0 | 36.7 | 46.7 | 43.3 | 36.7 |
| | Qwen-2.5-32B | 10.0 | 13.3 | 13.3 | 30.0 | 26.7 | 10.0 | 43.3 | 40.0 | 26.7 |
| | Qwen-3-32B-Think | 80.0 | 86.7 | 86.7 | 83.3 | 80.0 | 86.7 | 83.3 | 80.0 | 83.3 |
| | R1-distill-0528-8B | 76.7 | 80.0 | 80.0 | 83.3 | 80.0 | 80.0 | 80.0 | 80.0 | 80.0 |
| AIME 2025 (30) | Gemma-3-27B | 20.0 | 26.7 | 23.3 | 30.0 | 26.7 | 23.3 | 36.7 | 30.0 | 26.7 |
| | Qwen-2.5-32B | 10.0 | 13.3 | 20.0 | 40.0 | 40.0 | 23.3 | 46.7 | 33.3 | 36.7 |
| | Qwen-3-32B-Think | 73.3 | 80.0 | 83.3 | 83.3 | 73.3 | 83.3 | 80.0 | 76.7 | 80.0 |
| | R1-distill-0528-8B | 70.0 | 73.3 | 73.3 | 76.7 | 73.3 | 73.3 | 76.7 | 76.7 | 76.7 |
| BigCodeBench (1139) | Qwen-3-32B | 40.3 | 43.5 | 50.0 | 46.9 | 45.4 | 48.3 | 47.5 | 45.3 | 45.2 |
| | Deepseek-Coder | 25.0 | 28.8 | 38.2 | 37.5 | 35.5 | 32.6 | 39.3 | 35.1 | 33.3 |
| | Qwen-2.5-7B-Coder | 31.4 | 35.2 | 40.9 | 41.8 | 39.2 | 41.4 | 41.5 | 39.2 | 38.1 |
| IFEval (541) | Qwen-2.5-32B-Instruct | 78.6 | 80.6 | 82.1 | 86.0 | 84.7 | 81.0 | 86.0 | 83.2 | 80.6 |
| | Qwen-2.5-72B-Instruct | 82.4 | 84.5 | 85.2 | 88.4 | 89.1 | 84.8 | 88.0 | 87.2 | 83.9 |

Table 8: Ablation Study on Reward Formulation (Multiplication vs. Addition).

| Method | MMLU-P | MATH | GPQA | MBPP-P | IFEval | Avg. | IFBench | CJBench | RWBench | RMBench | JGBench |
|---|---|---|---|---|---|---|---|---|---|---|---|
| *Pointwise Evaluation* | | | | | | | | | | | |
| TIR-Judge-Distill 4B (Ours) | 58.7 | 81.9 | 45.8 | 64.1 | 78.9 | 65.9 | 65.8 | 59.9 | 76.6 | 71.9 | 66.7 |
| TIR-Judge-Zero 4B (Ours) | 62.5 | 87.3 | **54.7** | **64.8** | **79.8** | **69.8** | **65.9** | 61.5 | **77.3** | **72.8** | **70.4** |
| TIR-Judge-Distill 4B (Add.) | 62.5 | 73.4 | 46.9 | 63.5 | 74.2 | 64.1 | 65.5 | 59.3 | 73.7 | 68.6 | 60.6 |
| TIR-Judge-Zero 4B (Add.) | **63.5** | **89.8** | 48.5 | 62.0 | 68.5 | 66.5 | 63.0 | 61.1 | 76.0 | 63.4 | 68.6 |
| *Pairwise Evaluation* | | | | | | | | | | | |
| TIR-Judge-Distill 4B (Ours) | 69.0 | 88.7 | 54.8 | 60.6 | 83.6 | 71.3 | 73.7 | 69.8 | **87.7** | 78.0 | 70.5 |
| TIR-Judge-Zero 4B (Ours) | **75.0** | **93.3** | **61.7** | **67.3** | 84.5 | **76.3** | 70.3 | **70.8** | 86.7 | **80.8** | **73.7** |
| TIR-Judge-Distill 4B (Add.) | 56.5 | 85.4 | 43.9 | 56.5 | **85.5** | 65.6 | **74.5** | 66.3 | 84.6 | 77.4 | 65.3 |
| TIR-Judge-Zero 4B (Add.) | 69.5 | 91.9 | 52.1 | 63.5 | 81.2 | 71.6 | 72.0 | 68.7 | 85.4 | 80.0 | 72.0 |

Table 9: Position bias analysis. We report performance for A-B order, B-A order, and the average. Results indicate minimal variance for our method compared to baselines.

| Model / Setting | MMLU-P | MATH | GPQA | MBPP-P | IFEval | PPE Avg. | IFBench | CJBench | RWBench | RMBench |
|---|---|---|---|---|---|---|---|---|---|---|
| *TIR-Judge-Zero 4B* | | | | | | | | | | |
| A-B | 75.0 | 93.3 | 61.7 | 67.3 | 84.5 | 76.36 | 70.3 | 70.8 | 86.7 | 80.8 |
| B-A | 76.4 | 92.6 | 59.6 | 66.4 | 81.5 | 75.3 | 73.6 | 69.2 | 86.2 | 80.6 |
| Avg. | 75.7 | 93.0 | 60.6 | 66.9 | 83.0 | 75.8 | 72.0 | 70.0 | 86.4 | 80.7 |
| *TIR-Judge-Distill 4B* | | | | | | | | | | |
| A-B | 69.0 | 88.7 | 54.8 | 60.6 | 83.6 | 71.3 | 73.7 | 69.8 | 87.7 | 78.0 |
| B-A | 68.5 | 89.2 | 53.0 | 59.4 | 84.5 | 71.0 | 71.1 | 69.9 | 86.1 | 76.9 |
| Avg. | 68.7 | 89.0 | 53.9 | 60.0 | 84.1 | 71.1 | 72.4 | 69.9 | 86.9 | 77.5 |
| *Qwen3-4B-Tool* | | | | | | | | | | |
| A-B | 63.5 | 83.3 | 35.9 | 58.9 | 62.3 | 60.8 | 59.2 | 29.2 | 85.2 | 75.7 |
| B-A | 63.5 | 83.3 | 37.4 | 57.3 | 63.8 | 61.1 | 61.6 | 28.0 | 86.0 | 66.9 |
| Avg. | 63.5 | 83.3 | 36.7 | 58.1 | 63.0 | 60.9 | 60.4 | 28.6 | 85.6 | 71.3 |
| *TIR-Judge-Zero 8B* | | | | | | | | | | |
| A-B | 76.6 | 94.0 | 58.5 | 68.8 | 80.8 | 75.7 | 68.9 | 69.3 | 89.1 | 83.7 |
| B-A | 76.5 | 93.8 | 57.3 | 68.6 | 80.7 | 75.4 | 67.4 | 70.0 | 87.4 | 81.2 |
| Avg. | 76.6 | 93.9 | 57.9 | 68.7 | 80.8 | 75.6 | 68.1 | 69.6 | 88.3 | 82.4 |
| *TIR-Judge-Distill 8B* | | | | | | | | | | |
| A-B | 72.2 | 90.4 | 53.8 | 63.2 | 85.7 | 73.0 | 74.3 | 70.0 | 87.9 | 82.2 |
| B-A | 72.6 | 90.6 | 52.7 | 61.5 | 84.8 | 72.5 | 74.1 | 72.5 | 88.8 | 80.0 |
| Avg. | 72.4 | 90.5 | 53.3 | 62.3 | 85.2 | 72.8 | 74.2 | 71.2 | 88.3 | 81.1 |
| *Qwen3-8B-Tool* | | | | | | | | | | |
| A-B | 72.0 | 85.2 | 56.0 | 54.3 | 60.8 | 65.7 | 52.5 | 54.9 | 86.2 | 77.3 |
| B-A | 71.5 | 83.3 | 52.4 | 54.1 | 60.2 | 64.3 | 57.4 | 55.0 | 85.1 | 71.2 |
| Avg. | 71.8 | 84.3 | 54.2 | 54.2 | 60.5 | 65.0 | 55.0 | 55.0 | 85.7 | 74.3 |

From Table 7, we observe that `TIR-Judge` consistently delivers strong performance across model scales and judging formats, highlighting its robust generalization ability. These results demonstrate that `TIR-Judge` is not only effective but also readily transferable to diverse target tasks.

# G    ADDITIONAL STUDIES FOR `TIR-JUDGE`

**Different Reward Combinations.**    Table 8 shows the comparison of our proposed TIR-Judge (using multiplication reward formulation) against the addition-based reward formulation variant. We report results across both Pointwise and Pairwise settings. The result shows that using the addition form as the reward would lead to slightly worse performance.

**Performance of `TIR-Judge` with different orders.**    Table 9 shows that `TIR-Judge` exhibits small positional discrepancy (typically <1%, at most  2%), while the backbone Qwen3 models can have a relatively higher variance (up to 9%). This confirms that our training procedure effectively mitigates position bias.

**Performance of `TIR-Judge` with different response length.**    To evaluate the verbosity bias, we report the accuracy separately for cases where the chosen is longer or shorter than the response. The results show minimal difference between the two categories, and in some cases, `TIR-Judge` further reduces the verbosity gap observed in the Qwen3 backbone. This indicates that verbosity bias is well controlled.

Table 10: Accuracy Comparison Split by Response Length. We report the judge's accuracy when the ground-truth chosen response is longer than the rejected one versus when it is shorter (i.e., the rejected response is longer).

| Model | Acc. (Chosen > Rejected) | Acc. (Rejected > Chosen) |
|---|---|---|
| **TIR−Judge-Distill 4B** | 78.58 | 76.27 |
| **TIR−Judge-Zero 4B** | 80.67 | 79.27 |
| Qwen-3-4B-Tool (Backbone) | 65.79 | 71.20 |
| **TIR−Judge-Distill 8B** | 80.97 | 79.18 |
| **TIR−Judge-Zero 8B** | 77.73 | 77.98 |
| Qwen-3-8B-Tool (Backbone) | 72.61 | 72.25 |

Table 11: Comparison of Training Costs. **Left:** GPU wall-clock time breakdown. **Right:** Estimated financial cost including compute and API fees.

| Stage | TIR−Judge-Zero | TIR−Judge-Distill |
|---|---|---|
| SFT | 2.5h | 1.0h |
| RS (Rejection Sampling) | 3.5h | 1.5h |
| RL (Reinforcement Learning) | 23.0h | 8.5h |
| **Total Time** | **29.0h** | **11.0h** |

| Component | TIR−Judge-Zero | TIR−Judge-Distill |
|---|---|---|
| Compute Cost (8×H100) | ∼ $690 | ∼ $210 |
| Teacher API Cost | $0 | ∼ $130 |
| **Total Cost** | **∼ $690** | **∼ $340** |

## H    COST ANALYSIS OF TIR−JUDGE

We provide a detailed breakdown of the computational and financial costs for training TIR−Judge-8B in Table 11. Experiments were conducted on 8 NVIDIA H100 GPUs. We estimate the total cost based on current market rates for H100 clusters and the official API pricing (Gemini-2.5) for generating the 10k distillation samples. While TIR−Judge-Zero is approximately 2× more expensive (∼ $690 vs. ∼ $340), we argue this trade-off is strategically valuable. It unlocks *autonomous self-improvement* by eliminating the dependency on teacher supervision, making it a critical solution for privacy-sensitive environments where access to frontier models is restricted. In practice, our recommendations are:

- **Use Distill** for domains where there are noticeable gaps (measured by held-out evaluation sets) between the teacher and student model, provided API costs are permitted. It is ideal for injecting specific capabilities—such as safety, IF, correct tool-call formats, and thinking structures—that the base model lacks.

- **Use Zero** when no superior teacher exists (e.g., improving SOTA models) or to avoid API dependencies due to privacy concerns. It is ideal for unlocking latent reasoning capabilities through self-exploration, effectively trading API costs for training compute.

## I    MORE CASE STUDIES ON FAILURE CASES

We illustrate a representative failure case of TIR−Judge in Table 12. In this example, the model incorrectly counts the number of list items. The highlighted erroneous code shows that TIR−Judge attempts to match numbered lists beginning with a digit followed by a capital letter, but the actual response uses the "**" Markdown pattern for list items, which the regex fails to capture. This error arises because (1) the model does not robustly enumerate all formatting variants when generating code, and (2) although the task can be solved either with or without tool use, the model fails to recognize that direct reasoning (i.e., counting the items without code) would have been simpler and more reliable.

Additional failure cases include generating incorrect code (e.g., referencing undefined variables) or misunderstanding the semantics of the prompt—for instance, when asked to check for the presence of a {keyword}, TIR−Judge incorrectly interprets the literal braces "{}" as part of the keyword.

Table 12: A case study from IFBench of a failed judgement generated by `TIR-Judge`-Zero 8B. Red text stands for the incorrect reasoning step.

| | |
|---|---|
| **User Prompt** | Hello, how to learn AI using only online courses and tutorials? Please provide the answer with a light-hearted and encouraging tone. Provide a response that contains exactly 150 words, structured in a numbered list with each item beginning with a capital letter and ending with a period, and the word 'AI' should appear at least 3 times. |
| **Response** | 1. **Start with the Basics:** Begin with introductory courses on platforms like Coursera, edX, or Udacity to grasp the fundamentals of AI.
2. **Choose Your Language:** Learn Python, the most popular language for AI, through tutorials on Codecademy or freeCodeCamp.
3. **Dive into Data:** Explore data science and machine learning courses, as they are the backbone of AI, on websites like DataCamp or Kaggle.
4. **Hands-On Practice:** Apply your knowledge by participating in AI challenges on Kaggle or building small projects using datasets.
5. **Join Communities:** Engage with AI enthusiasts on forums like Reddit's r/MachineLearning or AI-specific groups on LinkedIn.
6. **Stay Updated:** Follow AI blogs, podcasts, and YouTube channels to keep up with the latest trends and breakthroughs.
7. **Never Give Up:** Remember, learning AI is a marathon, not a sprint. Enjoy the journey and celebrate your progress!
Keep pushing forward, AI mastery is within your reach! |
| **Judge Trajectory by** `TIR-Judge`**-Zero 8B** | To evaluate the response, I will check:
1. Whether the response is exactly 150 words.
2. Whether it's structured in a numbered list with each item beginning with a capital letter and ending with a period.
3. Whether the word 'AI' appears at least 3 times.

```python
import re
# Check word count
word_count = len(response.split())
# Check structure: each item starts with capital letter and ends with period
items = re.findall(r'\\d+\\.\\s+[A-Z].*?\\.', response)
# Check occurrences of 'AI'
ai_count = response.lower().count('ai')
print(f"Word count: word_count")
print(f"Number of AI occurrences: ai_count")
print(f"Number of list items: len(items)")
```
```output
Word count: 147
Number of AI occurrences: 9
Number of list items: 0
```
<score>1</score> |

