# OpenReview forum: "Incentivizing Agentic Reasoning in LLM Judges via Tool-Integrated Reinforcement Learning"
_ICLR.cc/2026/Conference — ICLR 2026 Poster_

### Official Review · Reviewer_dbDX · 2025-10-31

**Soundness:** 3
**Presentation:** 4
**Contribution:** 3
**Rating:** 6
**Confidence:** 4

**Summary:**

Proposes TIR-Judge, a framework to train (finetune) LLM judges. Specifically focuses on training them how to use tools (Python executor) using reinforcement learning, across verifiable and non-verifiable domains. Evaluate TIR-Judge on multiple benchmarks and show significant improvements over baseline, with just a small model.

**Strengths:**

1. Introduces a simple but effective framework to improve LLM-as-a-judge using RL on relatively small models.
2. Good figures and explanations.
3. Detailed evaluations and ablation comparisons. Well structured.

**Weaknesses:**

1. “eward-guided best-of-N inference over datasets” typo
2. Analysis on the code it does in Python? How often errors (runtime, syntax)?
3. Given they propose a framework, it is hard to know how well the training recipe and setup works in other setups. How well does it work on smaller models? Different model families?

**Questions:**

1. In Findeis 2025’s work, it was found that some of the benchmarks are too easy out of the box. How does your RL scale with complexity? E.g. does a better model still see improvements (on harder tasks), or do we only see gains here since it’s “low hanging fruit”?
2. "For safety and general helpfulness prompts, a positive format reward is granted only if the model produces a valid output without invoking tools" -- What is the impact of this?
3. Impact of training duration?

---

> ### Author Response · Authors · 2025-11-23
> **Initial Response to Reviewer dbDX (Part 1)**
>
> We appreciate your valuable insights and the time you dedicated to evaluating our paper. Our responses to your comments are as follows.
>
> ***
> > W1: “eward-guided best-of-N inference over datasets” typo
>
> A: Thanks for pointing this out! We have updated our manuscript to fix this typo.
>
> ***
> > W2: Analysis on the code it does in Python? How often errors (runtime, syntax)?
>
> A: Thanks for this suggestions. We further provide analysis on the breakdown of the error types of code generated by Qwen-3-8B-Tool and TIR-Judge-8B including (1) incorrect tool-call formats; (2) code syntax failures; (3) code runtime failures. Here are the results:
>
> | Code Error | TIR-Judge-Zero 8B | TIR-Judge-Distill 8B | Qwen-3-Tool |
> | :--- | :---: | :---: | :---: |
> | **Syntax** | 1.20% | 3.65% | 4.95% |
> | **Runtime** | 0 | 0 | 0 |
> | **Format** | 0.17% | 0.14% | 0.97% |
>
> The results confirms that TIR-Judge's improvement is driven by better code generation (significantly lower syntax errors) and reasoning capabilities, rather than simply correcting formatting artifacts. We have included these results with additional discussions in Section 5.5.
>
> ***
> > W3: Given they propose a framework, it is hard to know how well the training recipe and setup works in other setups. How well does it work on smaller models? Different model families?
>
> A: We thank the reviewer for raising this important point regarding the framework's robustness. To address this, we conducted additional experiments using a different model family and size: **Llama-3.2-3B**. We also included the result of using Llama-3.1-8B as a reference point.
>
> | Setting | Method | PPE | IFbench | CodeJudge | RewardBench | RM-Bench | JudgeBench |
> | :--- | :--- | :--- | :--- | :--- | :--- | :--- | :--- |
> | **Pointwise** | Backbone (3B) | 26.1 | 2.59 | 10.62 | 40.2 | 9.3 | 2.3 |
> | | Distill (3B) | **53.9** | **47.3** | **42.4** | **66.7** | 54.0 | 48.1 |
> | | Zero (3B) | 52.6 | 42.3 | 40.3 | 66.6 | **55.3** | **49.5** |
> | **Pairwise** | Backbone (3B) | 41.3 | 16.9 | 24.7 | 52.4 | 36.8 | 18.8 |
> | | Distill (3B) | **55.8** | **51.6** | 46.5 | **75.3** | 57.4 | **54.6** |
> | | Zero (3B) | 54.1 | 43.0 | **48.5** | 73.3 | **59.7** | 54.3 |
> | *Reference* | *Larger Backbone (Llama-3.1-8B)* | 54.7 | 13.2 | 32.3 | 69.5 | 54.0 | 32.3 |
>
> From the result, we have the following findings:
> - These results demonstrate that our training recipe effectively transfers to the Llama model family (distinct from the Qwen models in our main paper). While the absolute performance is naturally lower than larger models due to the limited intrinsic reasoning capabilities of Llama-series models, the improvements over the backbone are consistent and significant.
> - Both variants of TIR-Judge with 3B parameters outperforms the larger Llama-3.1-8B base model on key benchmarks, including RewardBench (+3.8) and RM-Bench (+5.7).
> - We observe that TIR-judge-zero performs slightly worse than distill in some 3B metrics. This is attributable to the "cold start" problem in smaller models: the lower reasoning capability of the 3B model results in a lower success rate during the Rejection Sampling stage, generating fewer high-quality trajectories for the RL optimization compared to larger models (e.g., Qwen).
>
> ***
> > Q1: In Findeis 2025’s work, it was found that some of the benchmarks are too easy out of the box. How does your RL scale with complexity? E.g. does a better model still see improvements (on harder tasks), or do we only see gains here since it’s “low hanging fruit”?
>
> A:  Thanks for this insightful question. We answer your question as follows:
> - First, our evaluation tasks **goes beyond simple evaluation** such as Rewardbench and are more complex: For example, we include evaluation on MMLU-Pro (college-level) and GPQA (PhD-level) reasoning problems, and CodeJudgeBench involves code generation tasks for competitive programming, which are all non-trivial and much harder than Rewardbench.
> - TIR-Judge achieved notable performance gains on those **more challenging benchmarks**. For instance, on the 4B model, while we see a modest 1.8% gain on RewardBench, we observe a 71.9% improvement on GPQA (PhD-level) and 139% on CodeJudgeBench. This suggests that our RL method effectively incentivizes the complex reasoning chains required for difficult problems, rather than just optimizing for simple surface-level patterns.
>
> ***
> > Q2: "For safety and general helpfulness prompts, a positive format reward is granted only if the model produces a valid output without invoking tools" -- What is the impact of this?
>
> A: This is because we want to teach the model not to call unnecessary tools. Our assumption is that safety and general helpfulness prompts generally do not require precise calculations, the model should handle them using its core knowledge base. Our experimental results also found that this leads to a slightly better performance.

---

> ### Author Response · Authors · 2025-11-23
> **Initial Response to Reviewer dbDX (Part 2)**
>
> > Q3: Impact of training duration?
>
> A: Thanks for this insightful question, we present the impact of Training Duration on TIR-Judge-Zero 4B.
>
> ### Performance by Training Round (Iterations)
>
> | Performance (Pointwise) | PPE | IFbench | CodeJudgeBench | RewardBench | RM-Bench | JudgeBench |
> | :--- | :--- | :--- | :--- | :--- | :--- | :--- |
> | **Round 0** | 65.6 | 60.8 | 57.5 | 76.9 | 72.4 | 65.3 |
> | **Round 1** | 69.2 | 64.4 | 60.2 | 75.8 | 71.9 | 71.0 |
> | **Round 2** | 69.8 | 65.9 | 61.5 | 77.3 | 72.8 | 70.4 |
>
> | Performance (Pairwise) | PPE | IFbench | CodeJudgeBench | RewardBench | RM-Bench | JudgeBench |
> | :--- | :--- | :--- | :--- | :--- | :--- | :--- |
> | **Round 0** | 67.3 | 65.5 | 66.4 | 85.5 | 79.1 | 71.0 |
> | **Round 1** | 75.9 | 68.5 | 70.1 | 87.9 | 81.4 | 72.9 |
> | **Round 2** | 76.3 | 70.3 | 70.8 | 86.7 | 80.8 | 73.7 |
>
> ### Performance by Training Steps (Round 2)
>
> | Pointwise (Round 2) | PPE | IFbench | CodeJudgeBench | RewardBench | RM-Bench | JudgeBench |
> | :--- | :--- | :--- | :--- | :--- | :--- | :--- |
> | **120 steps** | 67.6 | 65.2 | 60.8 | 77.8 | 72.0 | 68.8 |
> | **240 steps** | 68.9 | 65.9 | 61.2 | 77.7 | 72.2 | 70.2 |
> | **360 steps** | 69.8 | 65.9 | 61.5 | 77.3 | 72.8 | 70.4 |
>
> | Pairwise (Round 2) | PPE | IFbench | CodeJudgeBench | RewardBench | RM-Bench | JudgeBench |
> | :--- | :--- | :--- | :--- | :--- | :--- | :--- |
> | **120 steps** | 75.2 | 69.4 | 69.5 | 87.0 | 80.5 | 73.1 |
> | **240 steps** | 75.9 | 71.2 | 70.0 | 86.4 | 80.3 | 74.0 |
> | **360 steps (Ours)** | 76.3 | 70.3 | 70.8 | 86.7 | 80.8 | 73.7 |
>
> **Conclusion:** The above tables clearly indicates that (1) The training is relatively stable and there are no sharp fluctuations across different training steps/rounds.
> (2) Both increasing training iterations (Rounds) and training steps contribute positively to the final benchmark accuracy, with the **Pairwise method at 360 steps in Round 2** achieving the most robust overall results.
> (3) We can train for less steps to strike a balance between efficiency and accuracy,
>
> ***
> Thank you for taking the time to review our work. Your feedback is invaluable, and we welcome any additional questions or comments you might have.

---

### Official Review · Reviewer_LE7M · 2025-11-01

**Soundness:** 2
**Presentation:** 2
**Contribution:** 2
**Rating:** 4
**Confidence:** 4

**Summary:**

This paper presents TIR-Judge, a framework that trains LLM-based judges to integrate Python code execution with reinforcement learning, achieving strong empirical results across six benchmarks with notable parameter efficiency (8B model matching 32B baselines). The work demonstrates experimental coverage including pointwise/pairwise/listwise evaluation formats and best-of-N validation, with generally clear presentation and detailed appendices. The authors present the first approach to co-optimize inference and tool use (executing Python code) for LLM evaluator learning via reinforcement learning. Researchers achieved up to 6.4% performance improvement on pointwise and 7.7% performance on pairwise compared to robust inference-based evaluators on six public benchmarks, and 96% of Claudet-Opus-4 performance on 8B parameter models. TIR-Judge-Zero performs on par with or superior to distillation-based versions with only pure reinforcement learning without distillation from the teacher model.

**Strengths:**

1. Defining clear and practical issues
The practical limitations of LLM evaluators (accurate calculation, failure to verify constraints) are clearly presented with concrete examples.
The importance of evaluators has been convincingly described throughout the model development pipeline (post-training, inference, and evaluation).

2. Comprehensive Experimental Design
Researchers systematically evaluated it on six different benchmarks. They demonstrate generalizability by addressing all three evaluation formats: Pointwise, pairwise, and listwise.
Best-of-N experiments validate its practicality in downstream applications.

3. Thorough ablation study
Authors systematically analyzed the data mixing effect (Figure 3), tool usage vs text-only (Figure 4a), and progress of iterative RL (Figure 5).

4. Efficiency and Scalability
Parametric Efficiency: 8B model showed competitive performance with 32B model.
Computational Efficiency: We demonstrate no additional inference time overhead despite tool integration (Figure 6).

5. Efforts for Reproducibility
Addendum provided detailed prompt templates, dataset configurations, hyperparameters, and more.
It specified data pollution prevention efforts through 8-gram deduplication.

**Weaknesses:**

1. Problem: The key components of this paper (tool-integrated reasoning, RL-based judgments, and projection sampling) are all techniques covered in previous studies. Combining them together is meaningful, but there is a lack of theoretical analysis or deep insight into why this particular combination is effective.

Evidence: Feng et al. (2025), Li et al. (2025a), cited by Related Work, have already learned TIR as RL, while Chen et al. (2025b), Whitehouse et al. (2025) have dealt with RL-based judges. This paper is a combination of these, but does not explain why applying TIR to judges is essentially different from applying to policy models.


2. Lack of statistical rigor (Major)
Problem: All experimental results were reported as single run point estimates, and no confidence interval, standard deviation, or statistical significance tests were presented at all.

Evidence: All figures in Table 1 are shown up to a single decimal point, and there is no variance or p-value for multiple random seeds. In particular, we do not know if the performance difference between TIR-Judge-Zero and Distill (e.g., a 3.7% difference in 4B pointwise) is statistically significant.
Impact: The credibility of one of the main arguments, the conclusion that "distillation is unnecessary," is diminished.


3. Unfair comparison and baseline setting issues (Major)
Problem 1: The results of some strong baseline models (J1, GPT-4o, Claude 3.5, etc.) were cited in the original paper, and it is unclear whether they used exactly the same evaluation protocol and data preprocessing.
Problem 2: The "Tool-augmented" baseline (Qwen3-Tool, Gemini-2.5-Flash-Tool) seems to have induced tool use simply by prompting; for a fair comparison, they should have been fine-tuned with the same amount of tool use data.

Evidence: In Section 5.1, we said "For baseline, we consider... Qwen-3 (Team, 2025) (our backbone) that use the same tool as TIR-Judge," but they didn't specify how to learn. If you look at the prompt in Appendix B, it simply instructs you to "You may use Python code."
Impact: It's possible that the actual benefits of tool integration have been overestimated.



4. Lack of consistency in TIR-Judge-Zero's claim of excellence (Moderate)
Problem: While the paper emphasizes that TIR-Judge-Zero is effective without distillation, the real-world results are mixed with benchmarks and scales.

Evidence:
4B pairwise: Zero excels on 3 of 6 benchmarks
8B pairwise: Zero is only good on 2 of 6 benchmarks
4B pointwise: Zero is excellent in 7 out of 10 subtasks
8B pointwise: Distill excels in 6 out of 10 subtasks (Table 4)
Impact: The claim that "distillation is unnecessary" is exaggerated and should be corrected to "competitive performance can be achieved without distillation under certain conditions".

**Questions:**

1. Methodology Related (Critical)
Q1: Please clarify what exactly sθ(x, y) of pointwise evaluation means in Section 4.2 Equation (2). Is it the value in the <score> tag generated by the model or is it the scalar output inside the model? If it is the former, what do you do if an error occurs while parsing the generated text?

Q2: TIR-Judge-Zero's iterative learning said it would prefer to "keep only one trajectory per each prompt, with the shortest response or least tool calls," but how do you decide when these two criteria conflict (e.g., long response, but with fewer tool calls)? Do you have priorities?

Q3: Why did you choose the multiplication form from the reward function of Equation (3)? Have you tried the addition form (R = α₁ Rc + α₂Rf + α ₃ Rt)? Didn't the multiplication form cause the sparse reward problem early in learning?


2 Experimental Design Related (Critical)
Q4: How many random seeds did the results in Table 1 run? TIR-Judge-Zero 4B pairwise recorded 75.0% on PPE, what is the standard deviation or confidence interval for this? Specifically, did you test whether the performance difference between TIR-Judge-Zero and Distill was statistically significant?

Q5: How did you implement the "Tool-augmented Judges" baseline (Qwen3-Tool, Gemini-2.5-Flash-Tool)? Did you simply add "you may use Python code" to the prompt, or did you do new-shot prompting with some examples of tool usage? How much does learning them with data like TIR-Judge improve the performance?

Q6: Pairwise evaluation said you used "single random ordering", how did you control position bias? It's standard practice to evaluate and average both A-B and B-A orders, so why didn't you do this?


3. Interpretation of Results (Important)
Q7: Table 4 shows a mixture of cases where TIR-Judge-Zero is superior to distilled versions and cases where it is not. Under what conditions do you find a pattern that is better for Zero and under what conditions is better for Distill? Based on this, what would you recommend to practitioners?

Q8: In Figure 5, will the performance improve further or will it converge if we continue with an additional RL round after TIR-Judge-Zero-RL-2. What is the optimal number of iterations?

Q9: TIR-Judge on the RewardBench2 (Table 2) is higher than Claude-Opus-4 on the Instruction Followings, but far behind on Safety. What do you think is causing this difference? Could tool use be rather detrimental to safety assessments?
6.4 Regarding calculation costs (Important)

Q10: How do I compare the total learning cost (SFT + RL) of TIR-Judge-Distill to the total learning cost (RL + RS + SFT of multiple rounds) of TIR-Judge-Zero in GPU time? If the Zero method requires more computation, which is more efficient compared to the cost of generating a 10k sample of distillation data?

Q11: How many samples did Reception sampling generate on average per each prompt, and what percentage of them produced the correct format and answer? How did this acceptance rate change as learning progressed?


5. Generalization and Limitations Related (Moderate)
Q12: How do I modify my framework to extend it to other tools (web search, calculator, database query, etc.) other than the Python executor? Is the current reward function or learning procedure still applicable to other tools?

Q13: Has it occurred during learning or evaluation to generate malicious or dangerous code (e.g. file system access, infinite loop)? How did you detect and process it?

< Additional feedback for improvement >
1. Must-have
Add statistical significance: Rerun the experiment with at least 3-5 random seeds, report the mean±standard deviation. For key arguments (for example, Zero > Distill), provide a paired t-test or bootstrap confidence interval.

Compare fair baselines: Reevaluate tool-augmented baselines by fine tuning them into the same learning data. Separate the actual contribution of RL by adding "TIR-Judge without RL (only SFT)" as at least ablation.

Position bias control: Perform Pairwise evaluation in both A-B and B-A directions and average; current single ordering can have a maximum bias of ±5%.

Add an experiment to compare the multiplication form and the addition form of the reward function ablation: expression (3). This will show the robustness of the method.

2 Strongly recommended improvements
Calculate Cost Analysis: Add a table comparing the total cost of learning in Distill vs Zero to GPU time and dollar amounts. This is important for practitioners to make decisions.
Add theoretical analysis: Add theoretical insights or analyses on why tool integration is particularly effective for judges and why iterative RL converges. There is currently a lack of explanation beyond "it works empirically".

Failed Case Analysis: Add a section to analyze when TIR-Judge still fails. For example, if you write a code but make an error due to incorrect logic, or if you need to use the tool but don't use it, and so on.
Data Efficiency Analysis: What percentage of 26k samples actually require tool use?

---

> ### Author Response · Authors · 2025-11-23
> **Initial Response to Reviewer LE7M (Part 1)**
>
> We thank the Reviewer for the exceptionally detailed and constructive review. We appreciate the recognition of our comprehensive experimental design and parameter efficiency. We have taken your feedback and address them below.
>
> ***
> > W1: Problem: The key components of this paper (tool-integrated reasoning, RL-based judgments, and projection sampling) are all techniques covered in previous studies. Combining them together is meaningful, but there is a lack of theoretical analysis or deep insight into why this particular combination is effective.
> Evidence: Feng et al. (2025), Li et al. (2025a), cited by Related Work, have already learned TIR as RL, while Chen et al. (2025b), Whitehouse et al. (2025) have dealt with RL-based judges. This paper is a combination of these, but does not explain why applying TIR to judges is essentially different from applying to policy models.
> >
> > Strongly recommended improvements 2: Add theoretical insights or analyses on why tool integration is particularly effective for judges and why iterative RL converges. There is currently a lack of explanation beyond "it works empirically".
>
> A: Thanks for raising this concern. While the individual components we use (tool-use / RL) have appeared in prior work, we do not view this as a lack of novelty. Techniques such as RL are widely used across the community, yet what differentiates those works is not the presence of RL alone, but **the problem setting, the motivation for each component, the construction of data, the design of the reward and training pipeline**. From this perspective, our method is not a superficial combination of existing ideas, but a targeted design built around the specific challenges of LLM judges.
>
> - **Why applying TIR to Judges is important?** In particular, we start by identifying the structural limitations of judgment tasks. We find that for reasoning-heavy tasks, small models often struggle *more* to verify a complex trajectory than to generate one. For example, Qwen-3-4B achieves 83.4 in IFEval (policy model), yet the accuracy drops to 65.5 for pairwise judgement in IFeval.  This gap exists because judges lack the internal computation to simulate the correctness of a response. Judges must perform **precise evaluation** (lines 46-47), which directly motivates integrating a Python code executor into the reasoning loop.
>
> -  **Why RL for Training judges?**: Our experiments show that naïvely adding tools at **inference time** leads to suboptimal and often incorrect tool behavior (lines 55 & 350). Simple prompting is therefore insufficient. RL is required to teach judges when and how to invoke tools in order to produce accurate, verifiable decisions across diverse tasks (lines 65-66). Furthermore, we also **identify two realistic deployment scenarios**—whether a strong teacher model is available or not—and design two variants accordingly (Distill and Zero). Each design decision in our framework is thus grounded in concrete limitations or real-world constraints rather than arbitrary technique mixing.
>
> - **Regarding iterative RL**: The procedure converges because each iteration systematically corrects the tool-use failure modes from the previous one via rejection sampling. Early rounds may under- or over-use tools, while rejection sampling filters for concise, correct trajectories; the next RL round then reinforces these trajectories. This produces a stable refinement loop of **better examples → better rollouts → even better examples**. Since judgment rewards are **deterministic** (correct vs. incorrect), this iterative process converges reliably in practice. We have added clarifications in the revision to make these intuitions more explicit.

---

> ### Author Response · Authors · 2025-11-23
> **Initial Response to Reviewer LE7M (Part 2)**
>
> > W2: Lack of statistical rigor (Major) Problem: All experimental results were reported as single run point estimates, and no confidence interval, standard deviation, or statistical significance tests were presented at all.
> >
> > W4: Lack of consistency in TIR-Judge-Zero's claim of excellence (Moderate) Problem: While the paper emphasizes that TIR-Judge-Zero is effective without distillation, the real-world results are mixed with benchmarks and scales.
> >
> > Q4: Experimental Design Related (Critical) How many random seeds did the results in Table 1 run? TIR-Judge-Zero 4B pairwise recorded 75.0% on PPE, what is the standard deviation or confidence interval for this? Specifically, did you test whether the performance difference between TIR-Judge-Zero and Distill was statistically significant?
> >
> > Additional Feedback 1: Must-have Add statistical significance: Rerun the experiment with at least 3-5 random seeds, report the mean±standard deviation. For key arguments (for example, Zero > Distill), provide a paired t-test or bootstrap confidence interval.
>
>
> A: Thank you for the question. We address your concerns in the follow two parts:
>
> **On the statistical rigor problem (W2, Q4):**
>
> Our main experiments use **greedy decoding** with temperature = 0, under which LLM inference is **entirely deterministic**. Changing random seeds has minimal effect on the model outputs, so variance and confidence intervals are simply not defined in this setting. Besides, **none of the baselines** we compare against report standard deviations in their papers either. Our reporting is fully consistent with established practice.
>
> Nevertheless, to address the reviewer’s concern, we additionally conducted 5-run experiments under a stochastic setting (temperature = 0.8) and computed the standard deviations for TIR-Judge and best baseline under the same size (RM-R1-Distill-7B for PPE and JudgeBench, RRM for IFbench and CodeJudgeBench, Think-RM for RWBench, RM-Bench). The results are as follows (* stands for p<0.05 and ** stands for p<0.01 comparing with the best baseline)
>
> | | PPE | IFBench | CJBench | RWBench | RMBench | JGBench |
> |:-|:-:|:-:|:-:|:-:|:-:|:-:|
> | **Pointwise** | | | | | | |
> | Best Baseline (Similar Size) | 61.0±1.0 | 56.6±1.3 | 54.9±1.6 | 79.0±1.4 | 69.0±0.9 | 64.7±0.8 |
> | TIR-Judge-Distill 4B | 66.3±1.2** | 65.8±0.9** | 59.7±0.9** | 76.3±0.6 | 72.0±0.8** | 65.8±0.7* |
> | TIR-Judge-Zero 4B | 69.5±1.4** | 65.8±1.6** | 61.3±0.7** | 77.0±0.5 | 72.4±0.9** | **70.3±1.4** ** |
> | TIR-Judge-Distill 8B | **70.8±0.9** ** | **68.2±1.3** ** | **62.2±0.6** ** | 81.2±1.5* | **76.8±1.4** ** | 68.5±1.0** |
> | TIR-Judge-Zero 8B | 70.3±0.9** | 67.1±0.8** | 60.6±0.5** | **81.4±0.8** * | 76.6±1.2** | 67.0±1.5* |
> | **Pairwise** | | | | | | |
> | Best Baseline (Similar Size) | 66.8±1.5 | 59.9±1.2 | 63.7±1.5 | 85.7±0.8 | 74.0±1.0 | 68.0±1.2 |
> | TIR-Judge-Distill 4B | 71.1±1.0** | 73.8±0.5** | 69.8±0.5** | 87.6±1.1* | 78.7±0.7** | 70.7±1.4* |
> | TIR-Judge-Zero 4B | **76.3±0.8** ** | 70.2±0.9** | **70.7±0.7** ** | 86.7±0.6 | 80.2±0.5** | **73.1±1.6** ** |
> | TIR-Judge-Distill 8B | 73.1±0.8** | **74.7±0.9** ** | 70.0±1.1** | 87.8±0.4** | 82.0±0.9** | 71.1±1.5** |
> | TIR-Judge-Zero 8B | 75.4±1.3** | 69.1±0.6** | 69.3±0.8** | **89.0±0.8** ** | **83.6±0.8** ** | 72.0±1.5** |
>
>
> These results show that (1) TIR-Judge is stable across runs under stochastic decoding, and (2) TIR-Judge remains statistically stronger with $p<0.05$ for majority of datasets than the best baseline with similar size on all benchmarks.
>
> We emphasize, however, that our primary evaluation setting is temperature = 0, which by definition should not yield variance.
>
> &nbsp;
>
> **On the claim regarding “distillation is unnecessary” (W2, W4, Q4):**
>
> We would like to emphasize that **our paper never claims that distillation is unnecessary**. TIR-Judge-Distill and TIR-Judge-Zero are **two complementary variants of our method** that tailored for different resource constraints. Our comparisons are aimed at contrasting both variants against existing baselines, not against each other. What we state (as shown in Line 356) is that, TIR-Judge-Zero achieves comparable or even better performance than the Distill variant. This demonstrates that when teacher-generated trajectories are unavailable, the Zero variant is a practical and label-efficient alternative, though requiring more training rounds. This is different from claiming that “distillation is unnecessary”.
> We revise the manuscript to explicitly state: "*TIR-Judge-Zero offers a viable alternative for data-scarce regimes, achieving competitive performance to distillation-based methods, albeit with additional overheads in training time.*" in section 5.2.

---

> ### Author Response · Authors · 2025-11-23
> **Initial Response to Reviewer LE7M (Part 3)**
>
> > W3: Unfair comparison and baseline setting issues (Major) Problem 1: The results of some strong baseline models (J1, GPT-4o, Claude 3.5, etc.) were cited in the original paper, and it is unclear whether they used exactly the same evaluation protocol and data preprocessing. Problem 2: The "Tool-augmented" baseline (Qwen3-Tool, Gemini-2.5-Flash-Tool) seems to have induced tool use simply by prompting; for a fair comparison, they should have been fine-tuned with the same amount of tool use data.
> >
> > Q5: How did you implement the "Tool-augmented Judges" baseline (Qwen3-Tool, Gemini-2.5-Flash-Tool)? Did you simply add "you may use Python code" to the prompt, or did you do new-shot prompting with some examples of tool usage? How much does learning them with data like TIR-Judge improve the performance?
> >
> > Must-have Add 2: Compare fair baselines: Reevaluate tool-augmented baselines by fine tuning them into the same learning data. Separate the actual contribution of RL by adding "TIR-Judge without RL (only SFT)" as at least ablation.
>
> A: Thank you for raising these concerns. We address these issues below.
>
> **On the fairness of baseline results (J1, GPT-4o, Claude-3.5, etc.):**
>
> All benchmarks used in our evaluation are publicly released and well-established with **fixed evaluation protocols**. We do not perform prompt-level or response-level filtering of the test data.
> For proprietary models where access is restricted (e.g., J1, Claude-3.5), we cite officially reported numbers – a standard practice in the field. However, for all available open-weight baselines (RRM, RM-R1, Think-RM, AgentRM), we reproduced the results using our exact pipeline to guarantee strict alignment with our experimental setting.
>
> **On the tool-augmented baselines (Qwen3-Tool, Gemini-2.5-Flash-Tool):**
>
> For the tool-augmented baselines, **Gemini-2.5-Flash** uses its built-in tool-use interface, and for **Qwen3** we attach the **same external tool** as TIR-Judge and use the **same prompt** format described in Appendix B. We **do not simply instruct it that “you may use python code”**. Instead, as shown in the prompt format, the prompt explicitly specifies the required format to call the python execution tool using \```python and ```output tags, and we also indicate the scenarios when tool-use is helpful (e.g., checking capitalization, counting words or letters, verifying math steps, running test cases). In practice, we examined the outputs and found that this prompt instruction is already sufficient for the models to produce tool calls in the correct format.
>
> The reason these prompted tool-augmented baselines achieve suboptimal performance is that, even with this structured setup, the models still do not call tools consistently at the appropriate times: they may overuse tools when unnecessary, fail to call tools when required, or generate incorrect code. We further experimented with few-shot demonstrations to guide the models on when and how to call tools, and present the results below:
>
> || zero-shot | two-shot |
> |:-:|:-:|:-:|
> | Qwen3-4B-Tool (pointwise) | 49.8 | **50.7** |
> | Qwen3-8B-Tool (pointwise) | **34.0** | 33.2 |
> | Qwen3-4B-Tool (pairwise) | 62.3 | **63.4** |
> | Qwen3-8B-Tool (pairwise) | **60.8** | 58.9 |
>
> The results indicate that few-shot prompting improve performance marginally and sometimes even hurts it. This happens because tool-use can be applied to many different scenarios, and few-shot examples cannot cover them well, causing overfitting and degraded performance.
>
> These findings together justify **the necessity of post-training for tool use**.
>
> **For the suggestion to fine-tune the tool-augmented baselines with the same data:**
>
> We would like to point out that finetuning **Gemini-2.5-Flash** is not feasible as it is a proprietary model with no weights released publicly. For **Qwen3**, we already report results for each stage in **Figure 5** (SFT, RL, and iterative RL for both TIR-Judge-Distill and TIR-Judge-Zero). These results show that TIR-Judge consistently outperforms baselines with SFT-only (either via rejection sampling or distillation from Gemini-2.5-pro) and RL-only when trained on the same data.
>
> ***
> > Q1: Methodology Related (Critical) Q1: Please clarify what exactly sθ(x, y) of pointwise evaluation means in Section 4.2 Equation (2). Is it the value in the [object Object] tag generated by the model or is it the scalar output inside the model? If it is the former, what do you do if an error occurs while parsing the generated text?
>
> A: The $s_\theta(x,y)$ refers to the scalar score produced by the judge model inside the \<score>...\</score> tags during pointwise evaluation. If the model output does not include valid \<score> tags, or if the content between the tags cannot be parsed into an integer, we treat the output as an invalid format and set $R_c = 0$ in this case. **We added one sentence in section 4.2 to avoid misunderstanding.**

---

> ### Author Response · Authors · 2025-11-23
> **Initial Response to Reviewer LE7M (Part 4)**
>
> > Q2: TIR-Judge-Zero's iterative learning said it would prefer to "keep only one trajectory per each prompt, with the shortest response or least tool calls," but how do you decide when these two criteria conflict (e.g., long response, but with fewer tool calls)? Do you have priorities?
>
> A: Thanks for raising this question. We prioritize trajectories with the fewest tool calls, since encouraging efficient tool usage is the primary objective. If multiple trajectories tie under this criterion, we then choose the one with the shortest trajectory length to further promote concise and efficient reasoning. **We have added more details in Section 4.3 to explain this in our revised manuscript.**
>
> ***
> > Q3: Why did you choose the multiplication form from the reward function of Equation (3)? Have you tried the addition form (R = α₁ Rc + α₂Rf + α ₃ Rt)? Didn't the multiplication form cause the sparse reward problem early in learning?
> >
> > Must-have Add 4: Add an experiment to compare the multiplication form and the addition form of the reward function ablation: expression (3). This will show the robustness of the method.
>
> A: Thanks for the question. Our intuition for the reward design is that we want to assign zero reward to rollouts with incorrect answers, regardless of whether their format or tool calls are correct. The correctness reward is the most important criterion, and with the multiplication form, any rollout with $R_c=0$ automatically receives zero final reward. This prevents the model from learning from rollouts that produce incorrect final answers, even if they happen to follow the format or generate error-free code.
>
> Per your suggestion, we conducted additional experiments to train the judge model with the additive form ($R=R_c+R_f+R_t)$, and present the results in the following:
>
> | | MMLU-P | MATH | GPQA | MBPP-P | IFEval | Avg. | IFBench | CJBench | RWBench | RMBench | JGBench |
> |:-:|:-:|:-:|:-:|:-:|:-:|:-:|:-:|:-:|:-:|:-:|:-:|
> | **Pointwise** |
> | TIR-Judge-Distill 4B (ours) | 58.7 | 81.9 | 45.8 | 64.1 | 78.9 | 65.9 | 65.8 | 59.9 | 76.6 | 71.9 | 66.7 |
> | TIR-Judge-Zero 4B (ours) | 62.5 | 87.3 | 54.7 | 64.8 | 79.8 | 69.8 | 65.9 | 61.5 | 77.3 | 72.8 | 70.4 |
> | TIR-Judge-Distill 4B (addition form) | 62.5 | 73.4 | 46.9 | 63.5 | 74.2 | 64.1 | 65.5 | 59.3 | 73.7 | 68.6 | 60.6 |
> | TIR-Judge-Zero 4B (addition form) | 63.5 | 89.8 | 48.5 | 62.0 | 68.5 | 66.5 | 63.0 | 61.1 | 76.0 | 63.4 | 68.6 |
> | **Pairwise** |
> | TIR-Judge-Distill 4B (ours) | 69.0 | 88.7 | 54.8 | 60.6 | 83.6 | 71.3 | 73.7 | 69.8 | 87.7 | 78.0 | 70.5 |
> | TIR-Judge-Zero 4B (ours) | 75.0 | 93.3 | 61.7 | 67.3 | 84.5 | 76.3 | 70.3 | 70.8 | 86.7 | 80.8 | 73.7 |
> | TIR-Judge-Distill 4B (addition form) | 56.5 | 85.4 | 43.9 | 56.5 | 85.5 | 65.6 | 74.5 | 66.3 | 84.6 | 77.4 | 65.3 |
> | TIR-Judge-Zero 4B (addition form) | 69.5 | 91.9 | 52.1 | 63.5 | 81.2 | 71.6 | 72.0 | 68.7 | 85.4 | 80.0 | 72.0 |
>
> The result shows that using the addition form as the reward would lead to slightly worse performance. **We have added this experiment to Appendix G in the revised manuscript.**

---

> ### Author Response · Authors · 2025-11-23
> **Initial Response to Reviewer LE7M (Part 5)**
>
> > Q6: Pairwise evaluation said you used "single random ordering", how did you control position bias? It's standard practice to evaluate and average both A-B and B-A orders, so why didn't you do this?
> >
> > Must-have Add 3: Position bias control: Perform Pairwise evaluation in both A-B and B-A directions and average; current single ordering can have a maximum bias of ±5%.
>
> A: Thank you for this insightful question. During training, the order of the chosen and rejected responses is randomized for every pair, giving equal probability for either response to appear first. This forces the model to learn **position-invariant** behavior and prevents it from relying on positional heuristics.
>
> For evaluation, we follow prior works (e.g. J1, RRM and RM-R1) and use a single random ordering of the pair, with JudgeBench as the only exception, where we follow the original paper and consider both A-B and B-A.
>
> Per your suggestion, we follow the setting used in the J1 paper and additionally evaluated TIR-Judge under both directions and reported the individual A-B, B-A, and averaged results:
>
> | | MMLU-P | MATH | GPQA | MBPP-P | IFEval | PPE Avg. | IFBench | CodeJudgeBench | RewardBench | RMBench |
> |:-:|:-:|:-:|:-:|:-:|:-:|:-:|:-:|:-:|:-:|:-:|
> | **TIR-Judge-Zero 4B** |
> | A-B | 75.0 | 93.3 | 61.7 | 67.3 | 84.5 | 76.3 | 70.3 | 70.8 | 86.7 | 80.8 |
> | B-A | 76.4 | 92.6 | 59.6 | 66.4 | 81.5 | 75.3 | 73.6 | 69.2 | 86.2 | 80.6 |
> | avg. | 75.7 | 93.0 | 60.6 | 66.9 | 83.0 | 75.8 | 72.0 | 70.0 | 86.4 | 80.7 |
> | **TIR-Judge-Distill 4B** |
> | A-B | 69.0 | 88.7 | 54.8 | 60.6 | 83.6 | 71.3 | 73.7 | 69.8 | 87.7 | 78.0 |
> | B-A | 68.5 | 89.2 | 53.0 | 59.4 | 84.5 | 71.0 | 71.1 | 69.9 | 86.1 | 76.9 |
> | avg. | 68.7 | 89.0 | 53.9 | 60.0 | 84.1 | 71.1 | 72.4 | 69.9 | 86.9 | 77.5 |
> | **Qwen3-4B-Tool** |
> | A-B | 63.5 | 83.3 | 35.9 | 58.9 | 62.3 | 60.8 | 59.2 | 29.2 | 85.2 | 75.7 |
> | B-A | 63.5 | 83.3 | 37.4 | 57.3 | 63.8 | 61.1 | 61.6 | 28.0 | 86.0 | 66.9 |
> | avg. | 63.5 | 83.3 | 36.7 | 58.1 | 63.0 | 60.9 | 60.4 | 28.6 | 85.6 | 71.3 |
> | **TIR-Judge-Zero 8B** |
> | A-B | 76.6 | 94.0 | 58.5 | 68.8 | 80.8 | 75.7 | 68.9 | 69.3 | 89.1 | 83.7 |
> | B-A | 76.5 | 93.8 | 57.3 | 68.6 | 80.7 | 75.4 | 67.4 | 70.0 | 87.4 | 81.2 |
> | avg. | 76.6 | 93.9 | 57.9 | 68.7 | 80.8 | 75.6 | 68.1 | 69.6 | 88.3 | 82.4 |
> | **TIR-Judge-Distill 8B** |
> | A-B | 72.2 | 90.4 | 53.8 | 63.2 | 85.7 | 73.0 | 74.3 | 70.0 | 87.9 | 82.2 |
> | B-A | 72.6 | 90.6 | 52.7 | 61.5 | 84.8 | 72.5 | 74.1 | 72.5 | 88.8 | 80.0 |
> | avg. | 72.4 | 90.5 | 53.3 | 62.3 | 85.2 | 72.8 | 74.2 | 71.2 | 88.3 | 81.1 |
> | **Qwen3-8B-Tool** |
> | A-B | 72.0 | 85.2 | 56.0 | 54.3 | 60.8 | 65.7 | 52.5 | 54.9 | 86.2 | 77.3 |
> | B-A | 71.5 | 83.3 | 52.4 | 54.1 | 60.2 | 64.3 | 57.4 | 55.0 | 85.1 | 71.2 |
> | avg. | 71.8 | 84.3 | 54.2 | 54.2 | 60.5 | 65.0 | 55.0 | 55.0 | 85.7 | 74.3 |
>
> The results show that TIR-Judge exhibits **very small positional discrepancy** (typically <1%, at most ~2%), while the backbone Qwen3 models can have a relatively higher variance (up to 9%). This confirms that our training procedure effectively mitigates position bias. We have added these analyses in **Appendix G**.
>
> ***
> > Q7: Interpretation of Results (Important) Table 4 shows a mixture of cases where TIR-Judge-Zero is superior to distilled versions and cases where it is not. Under what conditions do you find a pattern that is better for Zero and under what conditions is better for Distill? Based on this, what would you recommend to practitioners?
>
> A: We observe that TIR-Judge-Distill tends to outperform Zero on tasks where the teacher significantly exceeds the student's base capabilities, such as on instruction following and safety domains. For instance, our teacher (Gemini-2.5-Flash) scores 84.0 on PPE-IFEval and 96.5 on RewardBench-Safety, while the Qwen-3-8B backbone scores only 60.8 and 86.2. In these cases, distillation transfers external knowledge that self-bootstrapping cannot easily acquire.
> Based on this, our recommendations are as follows:
> - **Use Distill** for domains where there are noticeable gaps (which can be measured by held-out evaluation sets) between the teacher and student model, provided API costs are permitted. It is ideal for injecting specific capabilities, such as safety, IF, correct tool-call formats, and thinking structures, that the base model lacks.
> - **Use Zero** when no superior teacher exists (e.g., improving SOTA models) or to avoid API dependencies due to issues such as privacy concerns. It is ideal for unlocking latent reasoning capabilities through self-exploration, effectively trading API costs for training compute.
> **We provide additional discussions on Appendix H on the recommendation to practitioners.**

---

> ### Author Response · Authors · 2025-11-23
> **Initial Response to Reviewer LE7M (Part 6)**
>
> > Q8: In Figure 5, will the performance improve further or will it converge if we continue with an additional RL round after TIR-Judge-Zero-RL-2. What is the optimal number of iterations?
>
> A: We follow your suggestion to present the accuracy of TIR-Judge-4B with an additional RL Round. The result is listed as follows:
>
> | Performance (Pointwise) | PPE | IFbench | CodeJudgeBench | RewardBench | RM-Bench | JudgeBench |
> | :--- | :--- | :--- | :--- | :--- | :--- | :--- |
> | **Round 0** | 65.6 | 60.8 | 57.5 | 76.9 | 72.4 | 65.3 |
> | **Round 1** | 69.2 | 64.4 | 60.2 | 75.8 | 71.9 | 71.0 |
> | **Round 2** | 69.8 | 65.9 | 61.5 | 77.3 | 72.8 | 70.4 |
> | **Round 3** | 70.2 | 65.4 | 62.1 | 77.8 | 72.8 | 71.0 |
>
> | Performance (Pairwise) | PPE | IFbench | CodeJudgeBench | RewardBench | RM-Bench | JudgeBench |
> | :--- | :--- | :--- | :--- | :--- | :--- | :--- |
> | **Round 0** | 67.3 | 65.5 | 66.4 | 85.5 | 79.1 | 71.0 |
> | **Round 1** | 75.9 | 68.5 | 70.1 | 87.9 | 81.4 | 72.9 |
> | **Round 2** | 76.3 | 70.3 | 70.8 | 86.7 | 80.8 | 73.7 |
> | **Round 3** | 76.8 | 71.2 | 70.4 | 87.0 | 80.5 | 74.1 |
>
> Based on these results, the gain of model performance **becomes smaller** as we add more rounds. While Round 3 offers slight refinements, Round 2 appears to be the optimal balance between computational efficiency and performance maximization. In practice, we recommend monitoring a held-out validation set to dynamically detect these plateaus and determine the optimal stopping point.
>
> ***
> > Q9: TIR-Judge on the RewardBench2 (Table 2) is higher than Claude-Opus-4 on the Instruction Followings, but far behind on Safety. What do you think is causing this difference? Could tool use be rather detrimental to safety assessments?
>
> A: The performance gap stems from data distribution, not tool use. Our training prioritized reasoning, utilizing only 1,000 safety prompts (<4% of total), whereas Claude-Opus has likely undergone extensive safety alignment. Crucially, tool use is **not detrimental**: Table 5 confirms that TIR-Judge improves upon the Qwen backbone's safety scores. This gap can be bridged in practice by increasing the volume of safety-specific training data in both SFT and RL stage.
>
> > Q10: How do I compare the total learning cost (SFT + RL) of TIR-Judge-Distill to the total learning cost (RL + RS + SFT of multiple rounds) of TIR-Judge-Zero in GPU time? If the Zero method requires more computation, which is more efficient compared to the cost of generating a 10k sample of distillation data?
> >
> > Strongly recommended improvements 1: Add a table comparing the total cost of learning in Distill vs Zero to GPU time and dollar amounts. This is important for practitioners to make decisions.
>
>
> A: We provide a detailed breakdown of the computational and financial costs for training the 8B model variants. All experiments were conducted on an 8 $\times$ NVIDIA H100 GPU cluster.
>
> **1. GPU Time Comparison**
> The table below details the wall-clock time for each training stage. TIR-Judge-Zero requires approximately 2.6$\times$ more total compute time (27.5h vs. 10.5h) due to the extensive exploration required during the iterative RL phase without teacher guidance.
>
> | Stage | TIR-Judge-Zero | TIR-Judge-Distill |
> | :--- | :---: | :---: |
> | **SFT** | 1.0h | 0.5h |
> | **RS** | 3.5h | 1.5h |
> | **RL** | 23.0h | 8.5h |
> | **Total Time** | **27.5h** | **10.5h** |
>
> **2. Financial Cost Comparison**
> We estimate the total cost based on current market rates for H100 clusters and the official API pricing (Gemini-2.5) for generating the 10k distillation samples.
>
> | Cost Component | TIR-Judge-Zero | TIR-Judge-Distill |
> | :--- | :---: | :---: |
> | **Compute Cost** (8xH100) | ~$690 | ~$210 |
> | **Teacher API Cost** | $0 | ~$130 |
> | **Total Cost** | **~$690** | **~$340** |
>
> While TIR-Judge-Zero is approximately 2$\times$ more expensive (~$690 vs. ~$340), we argue this trade-off is strategically valuable. It unlocks **autonomous self-improvement** by eliminating the dependency on teacher supervision, making it a critical solution for privacy-sensitive environments where access to external frontier models is restricted. We provide additional discussions on **Appendix H** on the cost.
>
> ***
> > Q11: How many samples did Reception sampling generate on average per each prompt, and what percentage of them produced the correct format and answer? How did this acceptance rate change as learning progressed?
>
> A: We generate **8 samples per prompt** during rejection sampling. The success rate (responses with correct format and answer) increased from **62.1%** in the first round (using the initial RL checkpoint) to **65.5%** in the second round. This upward trend confirms that the model effectively improved its reasoning and formatting capabilities throughout the iterative RL process.

---

> ### Author Response · Authors · 2025-11-23
> **Initial Response to Reviewer LE7M (Part 7)**
>
> > Q12: Generalization and Limitations Related (Moderate) How do I modify my framework to extend it to other tools (web search, calculator, database query, etc.) other than the Python executor? Is the current reward function or learning procedure still applicable to other tools?
>
> A:  Thank you for this insightful question. Our framework is designed to be tool-agnostic. We address the extension to new tools (e.g., Web Search, SQL, Calculator) from the following perspectives:
> - **Infrastructure extensibility**: The system is modular by design: adding a new tool only requires implementing the functions to parse actions, execute action, and observation processing. This allows our model to easily incorporate many other tools as well, such as SQL executors, symbolic algebra engines, theorem provers, or web search APIs.
> - **Reward Design**: Since the reward evaluates the *outcome* and *format* rather than the specific tool trace, the RL process naturally reinforces whatever tool usage leads to the correct solution (e.g., reinforcing correct SQL syntax for database QA or correct query formulation for fact-checking).
> - **Training pipeline reuse**:  The recipe remains unchanged. SFT provides the "cold start" for tool syntax, while the RL loop optimizes usage. For tools not supported by our current teacher (Gemini), we can bootstrap training data using Rejection Sampling on other capable open-source models (e.g., Qwen-3-Next) to generate valid initial trajectories.
>
> ***
> > Q13: Has it occurred during learning or evaluation to generate malicious or dangerous code (e.g. file system access, infinite loop)? How did you detect and process it?
>
> A: Thanks for the question. We did not observe any malicious or dangerous code during training or evaluation. Our execution framework includes multiple safety layers to prevent such behavior:
>
> - **Environment Isolation**: As noted in the Ethics Statement, all code runs in a fully sandboxed environment with restricted filesystem access and no network connectivity.
>
> - **Resource Limits (Infinite Loops)**: Each code block has a strict execution timeout (e.g., 5 seconds). Infinite loops or overly expensive operations are automatically terminated and marked as failed.
>
> - **Static Analysis (Dangerous Imports / Operations)**: Before execution, we filter out code containing high-risk imports or system-level operations (e.g., subprocess, socket, os.system, os.kill).
>
> These measures ensure that any potentially dangerous code is intercepted before it can run.
>
> ***
> > Strongly recommended improvements 3: Add a section to analyze when TIR-Judge still fails. For example, if you write a code but make an error due to incorrect logic, or if you need to use the tool but don't use it, and so on.
>
> A: Please refer to Appendix I for details.
>
> ***
> > Strongly recommended improvements 4: What percentage of 26k samples actually require tool use?
>
> A: Around 68% of the 26k samples use the python code executor, where helpfulness and safety do not require tool use, and 85% of the remaining samples use the tool.
>
> ***
> Thank you again for your helpful review. We truly appreciate your feedback and are glad to address any further questions or engage in additional discussion.

---

### Official Review · Reviewer_sHW4 · 2025-11-03

**Soundness:** 3
**Presentation:** 3
**Contribution:** 3
**Rating:** 6
**Confidence:** 3

**Summary:**

The paper introduces TIR-Judge, a framework for training LLM-based judges that integrate tool execution (Python interpreter) within a reinforcement learning pipeline. Unlike prior text-only judges, TIR-Judge enables LLMs to generate, execute, and learn from code to perform verifiable evaluations.

Empirical results on six major benchmarks (e.g., PPE, IFBench, CodeJudgeBench, RewardBench) show consistent improvements over reasoning-based judges.

**Strengths:**

1. Novel integration of tool-use with reinforcement learning: The work goes beyond inference-time tool invocation by embedding code execution into the RL loop, allowing models to learn when and how to use tools effectively.
2. Strong empirical performance: Demonstrated improvements across multiple datasets and formats, outperforming larger models like 32B RRM on several metrics. Particularly impressive is the performance of TIR-Judge-Zero, which learns without teacher supervision.

**Weaknesses:**

1.Domain bias: Gains are largest in verifiable domains (math, code), while improvements in non-verifiable areas (helpfulness, safety) are marginal.
2. Generalization limitations: The framework is Python-specific; it is unclear how well the method would extend to multiple heterogeneous tools or symbolic engines.

**Questions:**

1. Could iterative RL amplify over-confidence or lead to reward hacking?
2. How is when not to call a tool learned? Is this behavior emergent from the RL reward or manually encoded through heuristics?
3. Would a multi-tool setup (e.g., Python + web search + symbolic solver) be stable under the same RL objective?

---

> ### Author Response · Authors · 2025-11-23
> **Initial Response to Reviewer sHW4 (Part 1)**
>
> We thank the reviewer for your careful assessment and insightful comments on our paper. Our responses to your suggestions are provided below.
>
> ***
> > W1: Domain bias: Gains are largest in verifiable domains (math, code), while improvements in non-verifiable areas (helpfulness, safety) are marginal.
>
> A: Thanks for the thoughtful observation. The performance pattern is indeed expected, and we clarify the underlying reasons:
>
> 1. **Benchmarks in non-verifiable domains are already close to saturation**: Many helpfulness and safety benchmarks do not require deep reasoning to achieve high scores. For example, frozen Qwen-3-4B and Qwen-3-8B models already reach 92.7 and 93.3 on RewardBench-Chat, leaving limited headroom for improvement.
>
> 2. **Our method is explicitly designed for verifiable reasoning:** TIR-Judge introduces tool-use with code execution, which is particularly beneficial for domains requiring precise reasoning. This is evident on PPE, a more challenging benchmark where even DeepSeek-R1 reaches only 76.5%. Our model achieves 76.3% with just 8B parameters, demonstrating strong efficiency in verifiable domains.
>
> 3. **Maintaining non-verifiable performance is nontrivial**: Our goal is not to maximize performance in non-verifiable areas, but to ensure that specializing for verifiable tasks does not degrade performance elsewhere. Figure 3 shows that training only on tool-use, reasoning prompts yields performance drops on non-verifiable areas yet TIR-Judge with all datasets balance between verifiable and nonverifiable domains.
>
> ***
> > W2: Generalization limitations: The framework is Python-specific; it is unclear how well the method would extend to multiple heterogeneous tools or symbolic engines.
> >
> > Q3: Would a multi-tool setup (e.g., Python + web search + symbolic solver) be stable under the same RL objective?
>
> A: Thanks for this insightful question. We answer your question from four perspectives:
> - **Infrastructure extensibility**: The system is modular by design: adding a new tool only requires implementing the functions to parse actions, execute action, and observation processing. This allows our model to easily incorporate many other tools as well, such as SQL executors, symbolic algebra engines, theorem provers, or web search APIs.
> - **Reward Design**: Since the reward evaluates the *outcome* and *format* rather than the specific tool trace, the RL process naturally reinforces whichever tool usage leads to the correct solution (e.g., reinforcing correct SQL syntax for database QA or correct query formulation for fact-checking).
> - **Training pipeline reuse**:  The recipe remains unchanged. SFT provides the "cold start" for tool syntax, while the RL loop optimizes usage. For tools not supported by our current teacher (Gemini), we can bootstrap training data using Rejection Sampling on other capable open-source models (e.g., Qwen-3-Next) to generate valid initial trajectories.
> - **The stability of the RL Training**: Instability typically occurs as the model calling tools endlessly or redundantly. We address this in two ways: (1) SFT with rejection sampling ensures the model understands tool syntax and appropriate tool selection before RL begins, and (2) during RL, we penalize redundant tool calls and filter out overlong trajectories. Together, these steps constrain the search space and promote stable, efficient tool use.

---

> ### Author Response · Authors · 2025-11-23
> **Initial Response to Reviewer sHW4 (Part 2)**
>
> > Q1: Could iterative RL amplify over-confidence or lead to reward hacking?
>
> A: Thanks for pointing this out. For evaluation, we do not observe the case for severe reward hacking where LLM judges favors specific types of response (e.g. from specific policy model), as our evaluation covers diverse datasets including different types of response generated from different policy models (for example, PPE contains responses from 20 selected LLMs such as Mistral-large-2402, command-r, and  phi-3-medium-4k-instruct, which do not exist in responses of our training data). If reward hacking were occurring, we would expect the judge to favor "hacked" responses that do not actually solve the task, leading to a degradation in downstream performance. However, as shown in Figure 5, we observe consistent gains.
>
> To further verify this, we conduct evaluation on best-of-N setting on policy models (mentioned in section 5.4) using different round of RL models of TIR-judge. The accuracy is shown as belows:
>
> | Model Variant | AIME 24 | AIME 25 | BigCodeBench | IFEval |
> | :--- | :---: | :---: | :---: | :---: |
> | **TIR-Judge-4B (Iter 0)** | 0.583 | 0.550 | 0.404 | 0.832 |
> | **TIR-Judge-4B (Iter 1)** | 0.583 | 0.670 | 0.425 | 0.864 |
> | **TIR-Judge-4B (Iter 2)** | 0.600 | 0.575 | 0.430 | 0.872 |
> | | | | | |
> | **TIR-Judge-8B (Iter 0)** | 0.600 | 0.575 | 0.409 | 0.831 |
> | **TIR-Judge-8B (Iter 1)** | 0.620 | 0.583 | 0.420 | 0.858 |
> | **TIR-Judge-8B (Iter 2)** | **0.633** | **0.600** | **0.428** | **0.870** |
>
> As shown in the table, performance improves monotonically across iterations, confirming that the judge is learning meaningful verification signals that benefits the policy model rather than hacking the reward metric.
>
> ***
> > Q2: How is when not to call a tool learned? Is this behavior emergent from the RL reward or manually encoded through heuristics?
>
> A: We appreciate the question. The decision of *when not to call a tool* is handled through a combination of reward shaping, data curation, and emergent RL behavior instead of simple ad-hoc heuristics. Specifically:
>
> - **Reward shaping during RL**: For tasks where tool use is unnecessary (e.g., general safety or helpfulness judgments), the reward explicitly penalizes tool calls. If the model invokes a tool when it shouldn’t, the rollout receives lower reward. This directly trains the policy to avoid irrelevant tool use.
>
> - **Balanced supervision**: The SFT dataset intentionally includes a mix of trajectories that use tools and trajectories that solve the task without tools. This teaches the model that tool use is optional and only beneficial in the appropriate contexts.
>
> - **Rejection sampling**: In rejection sampling, when the model can answer correctly without a tool, that trajectory is selected over tool-dependent ones, reinforcing minimal and context-appropriate tool invocation.
>
> To further optimize the tool-use, we can add additional rewards with regard to the number of tool calls in RL training to further penalize unnecessary tool calls.
>
> ***
> Thank you once again for your insightful review. We appreciate your feedback on our work. Feel free to let us know if you have any further questions, and we are happy to discuss further.

---

### Official Review · Reviewer_LHuR · 2025-11-13

**Soundness:** 3
**Presentation:** 3
**Contribution:** 3
**Rating:** 6
**Confidence:** 4

**Summary:**

This paper introduces TIR-Judge, an end-to-end RL framework for training LLM-based Judges that interleaves reasoning with Python code execution. Using this framework, the authors train several TIR-Judge models, which tend to outperform other LLM-based Judges without tools.

**Strengths:**

1. The paper is well-written, and the motivation and approach are clear. The figures and case study enhance readability.
2. The proposed method framework is effective, and the evaluation is comprehensive, spanning a variety of baselines and several relevant benchmarks.
3. The flexible judgement formats (e.g., pointwise, pairwise, and listwise) expand the functionality of the framework.

**Weaknesses:**

1. Prior work has identified several biases common in LLM-based judges, e.g., positional bias [1], verbosity bias [2], and self-preference bias [3]. Other work [4] has shown that training can increase the prevalence of such biases. Given this, this work should evaluate for bias before and after training.
2. It's unclear how the baseline tool-augmented judges (e.g., Qwen3-4B-Tool or Gemini-2.5-Flash-Tool) are evaluated. Are they expected to output tool calls in the same format as TIR-Judge, e.g., '''python...''', or do they use the native tool-calling features of the models? If it's the former, it's difficult to assess whether the improvement comes from learning how to make effective tool calls versus just learning how to make tool calls using the correct format. If it's the latter, it should be made clear in the paper.
3. The studies on efficiency could be better explained. Is code execution time factored in? Why are TIR-Judge-Zero 4B and TIR-Judge-Distill 4B 2x as efficient as Qwen-3-4B (their base model), yet TIR-Judge-Distill 8B has similar latency to Qwen-3-8B? Can you provide the average number of generated tokens for each?
4. (minor) Table 6 is confusing. It reports the number correct instead of accuracy, and the lack of precision makes comparisons difficult. Is this over a single run of 16 responses?
[1] https://arxiv.org/pdf/2406.07791
[2] https://arxiv.org/pdf/2310.10076
[3] https://arxiv.org/pdf/2410.21819
[4] https://openreview.net/pdf?id=JFTSZa2stt

**Questions:**

see weakness

---

> ### Author Response · Authors · 2025-11-23
> **Initial Response to Reviewer LHuR (Part 1)**
>
> Thank you for your thorough review and helpful remarks. Below, we address each of your suggestions in detail.
>
> ***
> > W1: Prior work has identified several biases common in LLM-based judges, e.g., positional bias [1], verbosity bias [2], and self-preference bias [3]. Other work [4] has shown that training can increase the prevalence of such biases. Given this, this work should evaluate for bias before and after training. [1] https://arxiv.org/pdf/2406.07791 [2] https://arxiv.org/pdf/2310.10076 [3] https://arxiv.org/pdf/2410.21819 [4] https://openreview.net/pdf?id=JFTSZa2stt
>
> A: Thanks for the comments. We address your concerns as follows:
>
> **For positional bias:** During training, the order of the chosen and rejected responses is randomized for every pair, giving each response an equal probability of appearing first. This helps to learn **position-invariant** behavior and mitigates it from relying on positional heuristics.
>
> For evaluation, we follow prior works and use a single random ordering of the pair, with JudgeBench as the only exception, where we follow the original paper and consider both A-B and B-A.
>
> Per your suggestion, we follow the setting used in the J1 paper and additionally evaluated TIR-Judge under both directions and reported the individual A-B, B-A, and averaged results:
>
> | | MMLU-P | MATH | GPQA | MBPP-P | IFEval | Avg. | IFBench | CJBench | RWBench | RMBench |
> |:-:|:-:|:-:|:-:|:-:|:-:|:-:|:-:|:-:|:-:|:-:|
> | **TIR-Judge-Zero 4B** |
> | A-B | 75.0 | 93.3 | 61.7 | 67.3 | 84.5 | 76.36 | 70.3 | 70.8 | 86.7 | 80.8 |
> | B-A | 76.4 | 92.6 | 59.6 | 66.4 | 81.5 | 75.3 | 73.6 | 69.2 | 86.2 | 80.6 |
> | avg. | 75.7 | 93.0 | 60.6 | 66.9 | 83.0 | 75.8 | 72.0 | 70.0 | 86.4 | 80.7 |
> | **TIR-Judge-Distill 4B** |
> | A-B | 69.0 | 88.7 | 54.8 | 60.6 | 83.6 | 71.3 | 73.7 | 69.8 | 87.7 | 78.0 |
> | B-A | 68.5 | 89.2 | 53.0 | 59.4 | 84.5 | 71.0 | 71.1 | 69.9 | 86.1 | 76.9 |
> | avg. | 68.7 | 89.0 | 53.9 | 60.0 | 84.1 | 71.1 | 72.4 | 69.9 | 86.9 | 77.5 |
> | **Qwen3-4B-Tool** |
> | A-B | 63.5 | 83.3 | 35.9 | 58.9 | 62.3 | 60.8 | 59.2 | 29.2 | 85.2 | 75.7 |
> | B-A | 63.5 | 83.3 | 37.4 | 57.3 | 63.8 | 61.1 | 61.6 | 28.0 | 86.0 | 66.9 |
> | avg. | 63.5 | 83.3 | 36.7 | 58.1 | 63.0 | 60.9 | 60.4 | 28.6 | 85.6 | 71.3 |
> | **TIR-Judge-Zero 8B** |
> | A-B | 76.6 | 94.0 | 58.5 | 68.8 | 80.8 | 75.7 | 68.9 | 69.3 | 89.1 | 83.7 |
> | B-A | 76.5 | 93.8 | 57.3 | 68.6 | 80.7 | 75.4 | 67.4 | 70.0 | 87.4 | 81.2 |
> | avg. | 76.6 | 93.9 | 57.9 | 68.7 | 80.8 | 75.6 | 68.1 | 69.6 | 88.3 | 82.4 |
> | **TIR-Judge-Distill 8B** |
> | A-B | 72.2 | 90.4 | 53.8 | 63.2 | 85.7 | 73.0 | 74.3 | 70.0 | 87.9 | 82.2 |
> | B-A | 72.6 | 90.6 | 52.7 | 61.5 | 84.8 | 72.5 | 74.1 | 72.5 | 88.8 | 80.0 |
> | avg. | 72.4 | 90.5 | 53.3 | 62.3 | 85.2 | 72.8 | 74.2 | 71.2 | 88.3 | 81.1 |
> | **Qwen3-8B-Tool** |
> | A-B | 72.0 | 85.2 | 56.0 | 54.3 | 60.8 | 65.7 | 52.5 | 54.9 | 86.2 | 77.3 |
> | B-A | 71.5 | 83.3 | 52.4 | 54.1 | 60.2 | 64.3 | 57.4 | 55.0 | 85.1 | 71.2 |
> | avg. | 71.8 | 84.3 | 54.2 | 54.2 | 60.5 | 65.0 | 55.0 | 55.0 | 85.7 | 74.3 |
>
> The results show that TIR-Judge exhibits **very small positional discrepancy** (typically <1%, at most ~2%), while the backbone Qwen3 models can have a relatively higher variance (up to 9%). This confirms that our training procedure effectively mitigates position bias.
>
> **For verbosity bias:** To evaluate the verbosity bias, we report the accuracy separately for cases where the chosen response is longer vs. shorter:
>
> | | Accuracy when chosen is longer | Accuracy when rejected is longer |
> |:-:|:-:|:-:|
> | TIR-Judge-Distill 4B | 78.58 | 76.27 |
> | TIR-Judge-Zero 4B | 80.67 | 79.27 |
> | Qwen-3-4B-Tool (backbone) | 65.79 | 71.2 |
> | TIR-Judge-Distill 8B | 80.97 | 79.18 |
> | TIR-Judge-Zero 8B | 77.73 | 77.98 |
> | Qwen-3-8B-Tool (backbone) | 72.61 | 72.25 |
>
> The results show minimal difference between the two categories, and in some cases TIR-Judge further reduces the verbosity gap observed in the Qwen3 backbone. This indicates that verbosity bias is well controlled.
>
> **For self-preference bias:** The chances of self-preference bias in our setting is small because **none of the evaluation benchmarks contains responses generated by our backbone models (Qwen-3-4B/8B)**:
> - **Temporal Mismatch**: Qwen-3-4B/8B were released in May 2025, after PPE, RewardBench, RM-Bench, JudgeBench, and IFBench were created.
> - **Model Disparity**: For newer benchmarks, RewardBench2 explicitly excludes Qwen-3 (Table 6, original paper). CodeJudgeBench uses distinct models (Claude-4, Gemini-2.5) or significantly larger variants (Qwen-3-235B), preventing direct self-bias from the 4B/8B judge.
>
> We have added the above experimental results to **Appendix G** in the revised version.

---

> ### Author Response · Authors · 2025-11-23
> **Initial Response to Reviewer LHuR (Part 2)**
>
> > W2: It's unclear how the baseline tool-augmented judges (e.g., Qwen3-4B-Tool or Gemini-2.5-Flash-Tool) are evaluated. Are they expected to output tool calls in the same format as TIR-Judge, e.g., '''python...''', or do they use the native tool-calling features of the models? If it's the former, it's difficult to assess whether the improvement comes from learning how to make effective tool calls versus just learning how to make tool calls using the correct format. If it's the latter, it should be made clear in the paper.
>
> A: Thanks for raising this question.
>
> **For Gemini-2.5-Flash-Tool**, we use its native code execution tool (https://ai.google.dev/gemini-api/docs/code-execution). This ensures that Gemini uses its built-in mechanism for tool execution rather than adapting to TIR-Judge’s formatting style. We also note that these implementation details are explicitly documented in **Appendix E** (“Implementation details for baselines”).
>
> **For Qwen-4B/8B-Tool**, we do not use the Qwen native “function_call” setting mentioned in https://qwen.readthedocs.io/en/latest/framework/function_call.html, as that setting requires pre-defined function schemas and does not allow the model to freely write its own code. Instead, our setup asks Qwen to directly write executable Python code inside \```python ... ``` blocks, and we run the code through an external executor.
>
> To further clarify whether improvements come from formatting or from better tool-use and reasoning, we break down code errors into (1) incorrect tool-call formats, (2) syntax errors, and (3) runtime failures. Below are the results for Qwen-3-8B-Tool and TIR-Judge-8B:
>
> | Code Error | TIR-Judge-Zero 8B | TIR-Judge-Distill 8B | Qwen-3-Tool |
> | :--- | :---: | :---: | :---: |
> | **Format** | 0.17% | 0.14% | 0.97% |
> | **Syntax** | 1.20% | 3.65% | 4.95% |
> | **Runtime** | 0 | 0 | 0 |
>
> The results show that format errors in the Qwen backbone are already negligible (<1%). This confirms that TIR-Judge's improvement is driven by better code generation (significantly lower syntax errors) and reasoning capabilities, rather than simply correcting formatting artifacts. We have included these results with additional discussions in **Section 5.5**.
>
> ***
> > W3: The studies on efficiency could be better explained. Is code execution time factored in? Why are TIR-Judge-Zero 4B and TIR-Judge-Distill 4B 2x as efficient as Qwen-3-4B (their base model), yet TIR-Judge-Distill 8B has similar latency to Qwen-3-8B? Can you provide the average number of generated tokens for each?
>
> A: Yes, code execution time is included in the latency measurement. We present the average number of generated tokens of TIR-Judge and Qwen-3 in the following table.
> The primary reason is that Qwen-3-4B generates relatively longer outputs on average, which increases decoding time. Instead, TIR-Judge significantly reduces unnecessary verbosity compared to the backbone, with the Zero variant producing the most concise reasoning.
>
> | Model | Response Length |
> | :--- | :---: |
> | **TIR-Judge-Distill 4B** | 1199.2 |
> | **TIR-Judge-Zero 4B** | 913.4 |
> | **Qwen-3-4B-Tool (Backbone)** | 2478.8 |
> | **TIR-Judge-Distill 8B** | 1328.1 |
> | **TIR-Judge-Zero 8B** | 889.2 |
> | **Qwen-3-8B-Tool (Backbone)** | 1353.1 |
>
> ***
> > W4: (minor) Table 6 is confusing. It reports the number correct instead of accuracy, and the lack of precision makes comparisons difficult. Is this over a single run of 16 responses?
>
> A: Thanks for pointing this out. This experiment is conducted under best-of-N setting, which means for each single prompt, the policy model generate 16 different responses, then:
> - Pass@1 is the average number of correct answers out of 16 generations.
> - MV@16 is the number of correctly answered questions under majority voting.
> - The remaining results are the number of correct answers selected using either TIR-Judge or baseline judges.
>
> We have updated the table to report **accuracy** instead of raw counts to avoid confusion. Please refer to the revised manuscript for details.
>
> ***
> Thank you for taking the time to review our work so carefully. Your feedback is invaluable, and we welcome any additional questions or comments you might have.

---

### Author Response · Authors · 2025-11-23
**Summary of Revisions**

We thank the reviewers for their constructive suggestions and expert insights, which have helped us significantly strengthen the manuscript. We are encouraged that the reviewers recognized TIR-Judge as a **novel framework with flexible design** (sHW4, dbDX, LHuR) that demonstrates **strong empirical performance** (sHW4, LE7M, LHuR) and **parameter efficiency** (LE7M, dbDX), specifically noting its ability to **learn without distillation** (sHW4, LE7M).

Based on your feedback, we have made the following updates to the paper:
1. We added some additional studies of TIR-Judge in **Appendix G**, including switching the best-of-N results from the number of correct examples to accuracy, ablation of reward designs, positional bias and verbosity bias analysis.

2. We added a cost analysis to train TIR-Judge in **Appendix H**.

3. We added the discussion of the failure case of TIR-Judge in **Appendix I**.

4. We added a breakdown analysis of code execution errors in **Section 5.5**.

5. We added a few clarification sentences in the main contents to avoid confusion.

---

### Author Response · Authors · 2025-11-26
**Gentle reminder to check our rebuttal**

Dear Reviewers,

Thank you again for your time and valuable feedback. This is a gentle reminder that we have posted our response and revisions. We kindly invite you to review our response and join the discussion if possible. We are happy to address any further questions. Thank you!

Best,

Authors

---

### Meta-Review · Area_Chair_aapV · 2026-01-08

**Summary:**

This paper proposes TIR-Judge, an RL framework to train LLM judges that integrate Python code execution for precise evaluations across verifiable and non-verifiable tasks. Key contributions include end-to-end RL training without distillation, support for diverse judgment formats, and strong gains on benchmarks, with an 8B model rivaling larger ones.

**Reviewer Concerns:**

Reviewers praised the novel tool-RL integration, solid experiments, and efficiency but raised issues on biases (positional, verbosity), baseline fairness, statistical rigor, and generalization to other tools/models. The rebuttal addressed most via new experiments on biases, reward ablations, code errors, costs, and position swaps, plus clarifications on claims like distillation's role.

Outstanding: deeper theoretical insights on why the combo works, and full multi-seed variance reporting.

**Reviewer Scores:**

Initial scores were 6,6,4,6. Post-rebuttal, expect the 4 to rise to 6 (major concerns met with data); others likely hold or bump to 7 given added analyses strengthening claims.

---

### Decision · Program_Chairs · 2026-01-26

Accept (Poster)